# ARTDECO: Towards Efficient and High-Fidelity On-the-Fly 3D Reconstruction with Structured Scene Representation

**Guanghao Li**[1,2,3*] **Kerui Ren**[1,4*] **Linning Xu**[1,5] **Zhewen Zheng**[1,6]
**Changjian Jiang**[1,7] **Xin Gao**[1,2] **Bo Dai**[8] **Jian Pu**[2†] **Mulin Yu**[1†] **Jiangmiao Pang**[1]

[1]Shanghai Artificial Intelligence Laboratory, [2]Fudan University, [3]Shanghai Innovation Institute,
[4]Shanghai Jiao Tong University, [5]The Chinese University of Hong Kong, [6]Carnegie Mellon University,
[7]Zhejiang University, [8]The University of Hong Kong

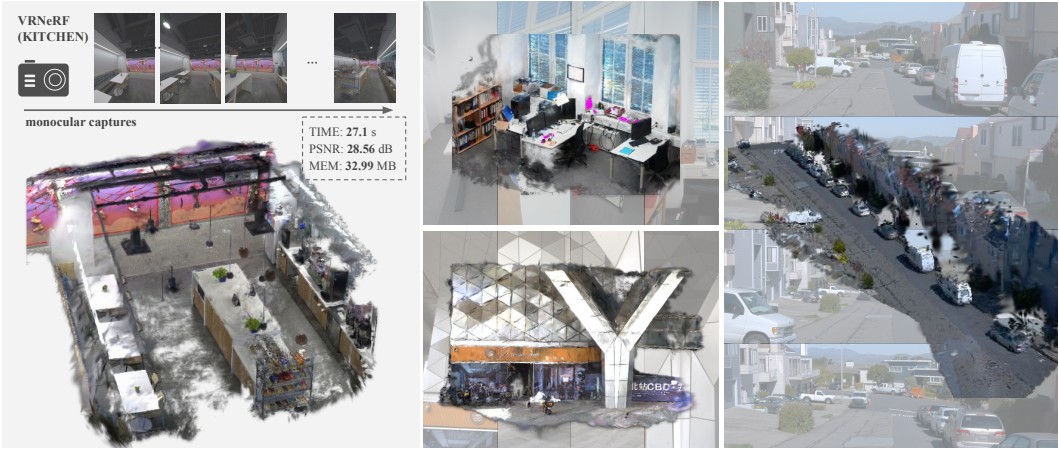

Figure 1: ARTDECO delivers high-fidelity, interactive 3D reconstruction from monocular images, combining efficiency with robustness across indoor and outdoor scenes.

## Abstract

On-the-fly 3D reconstruction from monocular image sequences is a long-standing challenge in computer vision, critical for applications such as real-to-sim, AR/VR, and robotics. Existing methods face a major tradeoff: per-scene optimization yields high fidelity but is computationally expensive, whereas feed-forward foundation models enable real-time inference but struggle with accuracy and robustness. In this work, we propose ARTDECO, a unified framework that combines the efficiency of feed-forward models with the reliability of SLAM-based pipelines. ARTDECO uses 3D foundation models for pose estimation and point prediction, coupled with a Gaussian decoder that transforms multi-scale features into structured 3D Gaussians. To sustain both fidelity and efficiency at scale, we design a hierarchical Gaussian representation with a LoD-aware rendering strategy, which improves rendering fidelity while reducing redundancy. Experiments on eight diverse indoor and outdoor benchmarks show that ARTDECO delivers interactive performance comparable to SLAM, robustness similar to feed-forward systems, and reconstruction quality close to per-scene optimization, providing a practical path toward on-the-fly digitization of real-world environments with both accurate geometry and high visual fidelity. Project page: **https://city-super.github.io/artdeco/**

---

*Equal contribution
†Corresponding author.

# 1 INTRODUCTION

High-fidelity 3D reconstruction from monocular image sequences is a long-standing goal in computer vision. Monocular data are relatively inexpensive, ubiquitous, and easy to capture, while accurate 3D scene representations are crucial for downstream applications such as embodied intelligence, AR/VR, or real-to-sim content creation. Recently, 3D Gaussian Splatting (Kerbl et al., 2023) has emerged as an efficient scene representation with strong empirical results. However, in monocular settings the lack of reliable geometric cues such as ambiguous scale, limited parallax, motion blur, and poor overlap makes it difficult to achieve accuracy, speed, and robustness at the same time. As a consequence, many existing systems optimize one objective at the expense of the others, limiting their practicality for online deployment.

Current 3DGS-based 3D reconstruction methods fall into two paradigms. Per-scene optimization methods (Matsuki et al., 2024; Huang et al., 2024b; Meuleman et al., 2025) rely on image poses estimated from Structure from Motion (SfM) or Simultaneous Localization and Mapping (SLAM), achieving high accuracy but at the cost of substantial computation, with robustness limited by the fragility of these pipelines. In contrast, recent feed-forward models (Tang et al., 2025; Ye et al., 2025; Jiang et al., 2025b; Zhang et al., 2025) leverage large-scale data to learn monocular priors and directly regress poses and Gaussian primitives with attention, enabling fast and robust inference across diverse scenes but with limited rendering fidelity and weak consistency. This tradeoff highlights the need for approaches that combine the efficiency of feed-forward models with the strengths of per-scene optimization methods to deliver accurate, robust, real-time reconstruction. Beyond the efficiency–accuracy tradeoff, 3DGS pipelines are also highly sensitive to scene scale. As the scene grows, the number of Gaussian primitives required for training and rendering increases rapidly, which reduces efficiency. Prior attempts to address this either apply post-hoc anchor-based pruning, which lowers computation but introduces boundary artifacts and increases memory cost, or add multi-scale Gaussians during training, which mitigates artifacts but lacks explicit structural organization. These limitations underscore the need for a principled and practical level-of-detail (LoD) mechanism in 3DGS.

**ARTDECO**[1] derives its name from a streamlined pipeline that unifies **A**ccurate localization, **R**obust recons**t**ruction, and **Deco**der-based rendering, with the aim of enabling on-the-fly 3D scene reconstruction and rendering. The core of ARTDECO is the goal of balancing real-time performance, accuracy, and robustness. It employs feed-forward models as data priors to reduce monocular ambiguities while maintaining the efficiency required for interactive use. To address the global inconsistency often seen in feed-forward approaches, ARTDECO integrates loop detection with lightweight bundle adjustment. Finally, a hierarchical semi-implicit Gaussian structure with LoD-aware densification provides level-of-detail control through a spatially sparse grid attached to Gaussians, which jointly decodes scale and rotation attributes from both grid and Gaussian features. This enables the system to scale effectively without excessive loss of fidelity or efficiency. Together, these components support practical real-time 3D reconstruction across diverse indoor and outdoor settings. Our main contributions can be summarized as follows:

- We present ARTDECO, an integrated system that unifies localization, reconstruction, and rendering into a single pipeline, designed to operate robustly across various environments.

- Notably, we incorporate *feed-forward foundation models* as modular components for pose estimation, loop closure detection, and dense point prediction. This integration improves localization accuracy and mapping stability while preserving efficiency.

- We further propose a *hierarchical semi-implicit Gaussian representation* with a LoD-aware densification strategy, enabling a principled trade-off between reconstruction fidelity and rendering efficiency, critical for large-scale, navigable environments.

- Extensive indoor and outdoor experiments show that ARTDECO achieves SLAM-level efficiency, feed-forward robustness, and near per-scene optimization quality, validating its effectiveness for practical on-the-fly 3D digitization.

---

[1]Beyond the acronym, the name also evokes the *Art Deco movement*, valued for structure, geometry, and clarity of form. This metaphor reflects our system's emphasis on structured scene representations.

## 2 RELATED WORK

### 2.1 MULTI-VIEW RECONSTRUCTION AND RENDERING

Neural Radiance Fields (NeRF) (Mildenhall et al., 2021) have attracted significant attention in novel view synthesis (NVS). NeRF and its variants (Barron et al., 2021; 2022; 2023; Zhang et al., 2020; Verbin et al., 2022) model continuous volumetric fields and achieve high-quality image synthesis. However, the reliance on expensive volume rendering and large networks results in long training times and hinders real-time applications. To address these limitations, several works (Müller et al., 2022; Xu et al., 2022; Sun et al., 2022) accelerate both training and rendering by introducing hybrid or explicit scene representations. Recently, 3D Gaussian Splatting (3DGS) (Kerbl et al., 2023) has shown remarkable progress in high-fidelity reconstruction and real-time rendering by representing scenes with anisotropic Gaussians and leveraging an efficient tile-based rasterizer. Following its introduction, research has largely focused on model compression (Fan et al., 2024; Chen et al., 2025), large-scale scene processing (Ren et al., 2025; Jiang et al., 2025c; Kerbl et al., 2024), and geometry reconstruction (Huang et al., 2024a; Yu et al., 2024c; Guédon et al., 2025; Yu et al., 2024a). Despite these advances in novel view synthesis (NVS), most methods assume access to accurate camera poses, typically estimated via Structure-from-Motion (SfM) (Schonberger & Frahm, 2016; Schönberger et al., 2016; Pan et al., 2024), which imposes considerable preprocessing costs for large-scale or in-the-wild captures. To alleviate this reliance, several works (Lin et al., 2021; Fu et al., 2024; Lin et al., 2025) propose joint optimization of camera poses and scene parameters. However, these approaches remain computationally intensive, are sensitive to wide-baseline settings, or still depend on costly post-refinement.

### 2.2 STREAMING PER-SCENE RECONSTRUCTION

Classical visual SLAM systems (Mur-Artal & Tardós, 2017; Engel et al., 2017; Teed et al., 2023; Li et al., 2025a; Chen et al., 2024a;b) provide online tracking, mapping, and loop closure, serving as the fundamental backbone for a wide range of computer vision (Yan et al., 2024b; Gao & Pu, 2025; Gao et al., 2025; Guo et al., 2025) and robotic tasks (Yan et al., 2025). However, these systems often rely on sparse or semi-dense geometric representations, which fall short in producing high-fidelity, photo-realistic maps required for immersive applications. To overcome this limitation, recent works integrate volumetric rendering techniques into SLAM to enable online NVS. Among these works, NeRF-based SLAM methods (Sucar et al., 2021; Zhu et al., 2022; Zhang et al., 2023; Li et al., 2026) exhibit photorealistic reconstruction but remain computationally expensive due to per-ray volumetric rendering. By contrast, 3DGS has gained traction for SLAM integration thanks to its explicit representation and efficient rendering. Utilizing the differential pipeline of 3DGS, some studies (Matsuki et al., 2024; Hu et al., 2024; Keetha et al., 2024; Yan et al., 2024a; Deng et al., 2024; Yu et al., 2025; Li et al., 2025b; Cheng et al., 2025a; Lin et al., 2025) directly propagate the gradient from the rendering loss to pose, while others (Yugay et al., 2024; Huang et al., 2024b; Peng et al., 2024; Ha et al., 2024; Peng et al., 2025; Wen et al., 2025; Sandström et al., 2025) leverage traditional SLAM Modules to provide accurate pose. However, in monocular setting these systems often struggle to balance robustness, accuracy, and runtime efficiency. Recently, On-the-fly NVS (Meuleman et al., 2025) has shown that GPU-friendly mini-bundle adjustment with incremental 3DGS updates can enable interactive reconstruction, though robustness on casual unposed videos remains limited.

### 2.3 FEED-FORWARD MODELS

Pretrained on large-scale datasets, recent feed-forward models (Wang et al., 2025a; 2024) reconstruct 3D scenes directly, avoiding per-scene optimization. These approaches can be divided into pose-aware and pose-free methods. Pose-aware models take images together with camera poses as input, enabling rapid reconstruction (Charatan et al., 2024; Chen et al., 2024c; Zhang et al., 2024; Jiang et al., 2025a). Pose-free models, in contrast, perform fully end-to-end reconstruction from raw images alone, typically representing scenes with either point maps (Wang et al., 2024; Leroy et al., 2024; Wang et al., 2025a;b; Murai et al., 2025) or 3DGS (Jiang et al., 2025b). Notably, these feed-forward methods offer robustness across diverse scenarios, remove the need for preprocessing, and allow fast inference suitable for interactive use. However, they generally achieve lower accuracy than per-scene optimized methods, and face challenges with maintaining global consistency, handling high-resolution inputs, and processing long sequences.

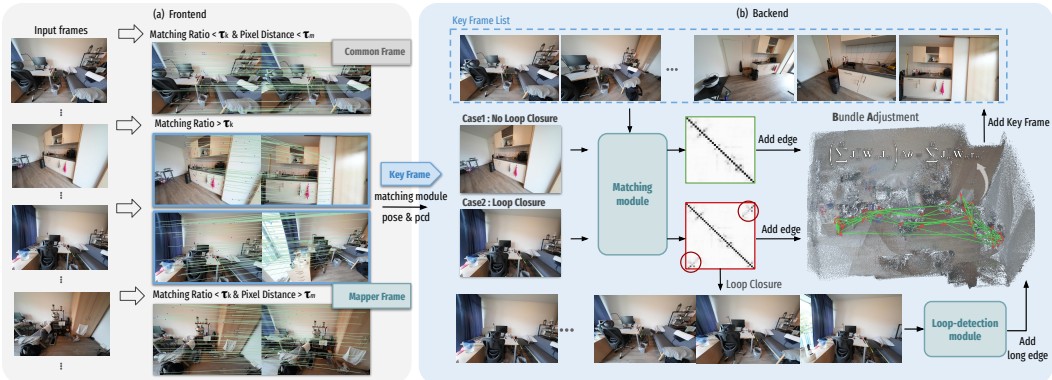

Figure 2: **Frontend and backend modules.** (a) *Frontend*: Images are captured from the scene and streamed into the frontend part. Each incoming frame is aligned with the latest keyframe using a *matching module* to compute pixel correspondences. Based on the correspondence ratio and pixel displacement, the frame is classified as a keyframe, a mapper frame, or a common frame. The selected frame, along with its pose and point cloud, is then passed to the back-end. (b) *Backend*: For each new keyframe, a *loop-detection module* evaluates its similarity with previous keyframes. If a loop is detected, the most relevant candidates are refined and connected in the factor graph; otherwise, the keyframe is linked only to recent frames. Finally, global pose optimization is performed with Gauss–Newton, and other frames are adjusted accordingly. We instantiate the matching module with MASt3R (Leroy et al., 2024) and the loop-detection module with $\pi^3$ (Wang et al., 2025b).

## 3 METHOD

We aim to recover a high-fidelity static 3D scene together with the corresponding camera poses from a monocular image sequence. Given a sequence of monocular RGB frames $\{\mathbf{I}_i\}_{i=1}^{N}$, with or without known camera intrinsics $\mathbf{K} \in \mathbb{R}^{3\times3}$, we estimate the camera poses $\{\mathbf{R}_i \mid \mathbf{t}_i\}_{i=1}^{N}$ associated with each image, as well as a set of Gaussian primitives $\{\mathcal{G}_j\}_{j=1}^{M}$ that compactly represent the 3D scene. By default, we assume the scene is static and rigid, and that all geometric information is inferred purely from monocular cues without external sensors.

As illustrated in Fig. 2 and 3, ARTDECO processes the sequence in a streaming SLAM-style pipeline consisting of three modules: *frontend*, *backend*, and *mapping*. (1) The frontend estimates relative poses and categorizes frames into common, mapping, or keyframes (Sec. 3.1). (2) The backend refines keyframe poses through loop closure and global bundle adjustment (Sec. 3.2). (3) Finally, image-wise pointmaps initialize 3D Gaussians, which are incrementally optimized in the mapping module (Sec. 3.3).

### 3.1 FRONTEND MODULE

For each input frame, the frontend estimates its pose relative to the latest keyframe and categorizes it as *common*, *mapping*, or *keyframe*. We assume a pinhole camera with fixed intrinsics and a shared optical center. If the focal length is unknown, it is initialized from the first $k_f$ GeoCalib (Veicht et al., 2024) estimates and jointly refined during pose estimation.

**Pose Estimation.** MASt3R (Leroy et al., 2024) serves as our *matching module*, a two-view reconstruction and matching prior, to improve camera tracking and focal length estimation. Following MASt3R-SLAM (Murai et al., 2025), we obtain frame-wise pointmaps, their confidence scores, and pixel correspondences between the current frame and the latest keyframe. The 3D points from the current frame are projected into the keyframe image plane, and the relative pose $\mathbf{T}_{KC} \in \mathrm{SIM}(3)$ is estimated by minimizing reprojection residuals with a Gauss–Newton solver. Since MASt3R predictions are less stable near object boundaries, we weight residuals by per-point uncertainty. For each point $\mathbf{x}_c$ in the current frame, we estimate a local covariance $\mathbf{\Sigma}_c \in \mathbb{R}^{3\times3}$ from neighbors within radius $\delta$. We then project $\mathbf{\Sigma}_c$ to the current keyframe's measurement space, which is used to filter out unreliable re-projection residuals. Besides, if the focal length is not provided, it is jointly optimized along with the relative pose. Further derivations are given in A.2.

**Keyframe Selection.** After pose estimation, each frame is categorized as a common frame, mapper frame, or keyframe. A *keyframe* is created when the number of valid correspondences with the latest keyframe falls below a threshold $\tau_k$, following standard SLAM practice. Keyframes are passed to the backend for pose refinement and to the mapping module for reconstruction. A *mapper frame* is selected when the frame provides sufficient parallax for reliable multi-view reconstruction. We compute the pixel displacement between the current frame and the latest keyframe; if the 70th percentile exceeds $\tau_m$, the frame is promoted to a mapper frame. Mapper frames are first processed by the backend to compute pointmap confidence and are then used in the mapping module to initialize new 3D Gaussians. A *common frame* does not meet either the keyframe or mapper criteria and is therefore used only to refine existing scene details, without introducing new structure; its role will be further elaborated in later sections.

## 3.2 BACKEND

The backend processes keyframes from the frontend to maintain a globally consistent scene and camera trajectory. For each incoming keyframe, it evaluates correlations with earlier ones, builds a *factor graph* over the most relevant candidates, and performs global optimization to enforce multi-view consistency. In addition, it estimates the confidence of keyframe and mapper pointmaps, which are later used to initialize 3D Gaussian in the mapping module.

**Loop Closure and Global Bundle Adjustment.** Given a new keyframe, the backend first updates the factor graph by connecting it to related frames, and then performs a PnP-based global bundle adjustment (BA) to refine poses, as illustrated in Fig. 2.(b). If a loop closure is detected, the current keyframe is linked to its three most relevant predecessors; otherwise, it is connected only to the latest keyframe. Loop detection is initially performed using the Aggregated Selective Match Kernel (ASMK). A loop is declared when a previous keyframe has an ASMK score above a threshold $\tau_{\text{loop}}$ and is at least $k_{lopp}$ keyframes apart. To increase robustness against weak correspondences and noisy inputs, we further leverage the 3D foundation model $\pi^3$ (Wang et al., 2025b) as our *loop-detection module*. Specifically, the current frame and the top $N_a$ candidates from ASMK are processed by $\pi^3$ to produce pointmaps, from which we select the three most geometrically consistent keyframes based on angular error following (Murai et al., 2025). These are then connected to the factor graph, yielding more reliable loop closures and reducing drift. More details can be found in A.9.

**Pointmap Confidence.** We estimate pointmap confidence using reprojection error rather than relying on the confidence values predicted by MASt3R. When a mapper frame or keyframe is processed, its pointmap is projected onto the $N_c$ previous keyframes with the highest ASMK scores. For each point, we compute the reprojection errors across the $N_c$ keyframes, average them to obtain $\bar{e}$, and define the confidence score as $C = 1$ if $\bar{e} \leq \varepsilon_c$, and $C = \frac{1}{\bar{e} - \varepsilon_c + 1}$ otherwise, where $\varepsilon_c$ is a predefined threshold. This reprojection-based confidence provides a more reliable measure of geometric consistency across frames.

## 3.3 MAPPING MODULE

The mapping module reconstructs the 3D Gaussian scene from incoming frames, their estimated poses, and pointmaps, as illustrated in Fig. 3. Unlike prior 3DGS-based SLAM methods that rely only on keyframes, we leverage *all* frames to maximize the use of captured information: keyframes and mapper frames introduce new Gaussians, while common frames refine existing ones. This design enriches both visual and geometric details, which are critical not only for accurate reconstruction but also for high-fidelity rendering. Moreover, we introduce a hierarchical Gaussian structure with LoD-aware control. LoD is essential for scalable scene modeling, particularly in large-scale, navigable spaces where SLAM-based applications require consistent detail at varying viewing distances. By combining dense supervision from all frame types with principled level-of-detail management, our mapping module improves fidelity of both reconstructed geometry and rendered views, while maintaining computational efficiency.

**Probabilistic Selection for 3D Gaussian Insertion.** When a mapper frame or keyframe arrives from the backend, we determine where to initialize new 3D Gaussians. To avoid redundancy, Gaussians are inserted only in regions that require refinement, rather than at every pixel, guided by image-level priors inspired by (Meuleman et al., 2025). We prioritize high-frequency regions and poorly reconstructed areas by computing an insertion probability at each pixel $(u, v)$ using the Laplacian of

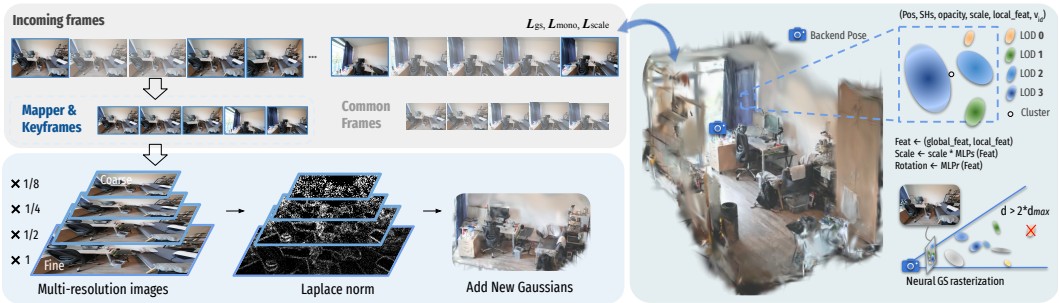

Figure 3: **Mapping process.** When a keyframe or mapper frame arrives from the backend, new Gaussians are added to the scene. Multi-resolution inputs are analyzed with the Laplacian of Gaussian (LoG) operator to identify regions that require refinement, and new Gaussians are initialized at the corresponding monocular depth positions in the current view. Common frames are not used to add Gaussians but contribute through gradient-based refinement. Each primitive stores position, spherical harmonics (SH), base scale, opacity, local feature, $d_{max}$, and voxel index $v_{id}$. For rendering, the $d_{max}$ attribute determines whether a Gaussian is included at a given viewing distance, enabling consistent level-of-detail control.

Gaussian (LoG) operator (Haralock & Shapiro, 1991):

$$P_a(u,v) = \max\Big( \min(\|\nabla^2(G_\sigma) * I(u,v)\|, 1) - \min(\|\nabla^2(G_\sigma) * \tilde{I}(u,v)\|, 1), 0\Big), \quad (1)$$

where $I$ and $\tilde{I}$ are the ground-truth and rendered images, and $G_\sigma$ is a Gaussian kernel with standard deviation $\sigma$. A new Gaussian is added when $P_a(u,v)$ exceeds the threshold $\tau_a$.

**Gaussian Primitive Initialization.** After identifying candidate pixels, we initialize the corresponding 3D Gaussians. Each Gaussian is parameterized by its center $\mu$, spherical harmonics (SH), opacity $\alpha$, base scale $S_b$, individual feature $f_l$, and voxel index $v_{id}$. The $\mu$ and SH0 are initialized from the pointmap and pixel color, while opacity is set to $0.2 \cdot C_{(u,v)}$ to down-weight low-confidence regions, where $C_{(u,v)}$ is the confidence score calculated in *backend*. Following (Meuleman et al., 2025; Wu et al., 2025; Yu et al., 2024b), the base scale at pixel $(u,v)$ is defined as:

$$S_b = \frac{d_i s'}{f}, \qquad s' = \frac{1}{2\sqrt{\min(\|(\nabla^2 G_\sigma) * I(u,v)\|, 1)}}, \quad (2)$$

where $d_i$ is the distance from the Gaussian center to the camera and $f$ is the focal length. Here, $s'$ represents an image-space scale, i.e., the expected distance to the nearest neighbor under a local 2D Poisson process of intensity $\min(\|(\nabla^2 G_\sigma) * I(u,v)\|, 1)$, (Clark & Evans, 1954). To ensure smoother reconstruction, we further refine scale and initialize rotation with two MLPs:

$$S = S_b \cdot \mathrm{MLP}_s(f_r \oplus f_l), \qquad R = \mathrm{MLP}_r(f_r \oplus f_l), \quad (3)$$

where $\oplus$ denotes concatenation, $f_l$ is an individual feature initialized as zero, and $f_r$ is a region feature encoding local voxel context. We voxelize the 3D space with cell size $\epsilon$; when a new Gaussian is added to a voxel, the corresponding voxel-wise feature is initialized as zero and indexed by $v_{id}$. This hybrid region–individual design promotes global consistency of the Gaussian field while preserving local distinctiveness.

**Levels of Detail Design.** To support smooth navigation in large 3D scenes, we organize Gaussians into multiple levels of detail (LoD). Each Gaussian is assigned a level $l \in \mathbb{N}^+$ with $l < L$, where level 0 denotes the finest resolution and level $L-1$ the coarsest. At initialization, a Gaussian at level $l$ corresponds to a patch of $2^{2l}$ pixels in the original image (e.g., level 0 corresponds to one pixel). We progressively downsample the input frame $L-1$ times and initialize Gaussians from both the downsampled and original images. All Gaussian parameters follow the initialization described earlier, except that (i) the base scale is weighted by $2^{2l}$, and (ii) each Gaussian is assigned a distance-dependent parameter $d_{max} = D \cdot 2^{2l}$, where $D$ is the distance from the Gaussian center to the camera. During rendering, a Gaussian is included if $d_r \leq d_{max}$, excluded if $d_r > 2d_{max}$, and smoothly faded out for $d_{max} < d_r \leq 2d_{max}$ by interpolating its opacity as $\alpha' = \alpha * (2d_{max} - d_r)/d_{max}$, where $\alpha$ represents the original opacity of the Gaussians. This distance-aware LoD design suppresses flickering and maintains stable rendering quality across scales while preserving efficiency.

**Training Strategy.** To balance efficiency and reconstruction quality, we adopt a staged training scheme. For streaming input, new Gaussians are initialized and the scene is optimized for $K$ iterations whenever a mapper frame or keyframe arrives, while common frames trigger only $K/2$ iterations without inserting new Gaussians. Following (Meuleman et al., 2025; Wu et al., 2025), training frames are sampled with a 0.2 probability from the current frame and 0.8 from past frames to mitigate local overfitting. After processing the sequence in a streaming manner, we run a global optimization over all frames, giving higher sampling probabilities to those with fewer historical updates. Finally, camera poses are optimized jointly with Gaussian parameters, with gradients on positions and rotations propagated to poses, consistent with common practice in on-the-fly reconstruction.

## 4 EXPERIMENTS

### 4.1 EXPERIMENTAL SETUP

**Datasets and Metrics.** We evaluate on diverse indoor and outdoor benchmarks. Indoor datasets include 11 TUM scenes (Sturm et al., 2012), 14 scenes from ScanNet++ (Yeshwanth et al., 2023), 8 scenes from VR-NeRF (Xu et al., 2023), and 6 scenes from ScanNet (Dai et al., 2017), with sequence lengths ranging from 32–5577 image frames. Outdoor datasets include 8 KITTI scenes (Geiger et al., 2013), 9 Waymo scenes (Sun et al., 2020) (both following S3PO-SLAM (Cheng et al., 2025a)), 5 scenes from Fast-livo2 (Zheng et al., 2024), and 1 scene from MatrixCity (Li et al., 2023), with lengths 200–1363 frames per trajectory. Reconstruction is evaluated with PSNR, SSIM (Wang et al., 2004), and LPIPS (Zhang et al., 2018); pose accuracy with Absolute Trajectory Error (ATE) RMSE; and system efficiency with FPS.

**Baselines.** We evaluate against two categories of state-of-the-art methods. For reconstruction quality, we consider 3D Gaussian Splatting approaches, including OnTheFly-NVS (Meuleman et al., 2025)), LongSplat (Lin et al., 2025), S3PO-GS (Cheng et al., 2025a), SEGS-SLAM (Wen et al., 2025), MonoGS (Matsuki et al., 2024)). For pose estimation, in addition to the aforementioned 3DGS-based SLAM methods, we benchmark against several state-of-the-art SLAM systems, including MASt3R-SLAM (Murai et al., 2025), DPV-SLAM (Lipson et al., 2024), DROID-SLAM (Teed & Deng, 2021), and Go-SLAM (Zhang et al., 2023). .

**Implementation Details.** Experiments are run on a desktop with an Intel Core i9-14900K CPU and NVIDIA RTX 4090 GPU. Following standard Novel View Synthesis practices, every 8th frame is held out for evaluation: these frames are excluded from mapping but their poses are estimated and optimized for evaluation. In the *frontend*, we set $\tau_k = \max(0.333 \cdot W, 30)$, where $W$ is the image width. In the *backend*, we use $N_a = \min(23, N_c)$ candidate keyframes, where $N_c$ is the number of available candidates. If $N_a < 11$, loop-detection modules are disabled, and the top three ASMK-scoring keyframes are directly selected to connect in the factor graph. For pointmap confidence, we fix $\varepsilon_c = 3$. During *mapping*, 3D Gaussians are organized into 4 LOD levels by setting $L = 4$.

### 4.2 COMPARISON

**Reconstruction Results Analysis.** Tab. 1 reports reconstruction results on eight indoor and outdoor benchmarks. Our system achieves state-of-the-art quality, particularly on challenging datasets such as TUM and ScanNet with structural complexity, motion blur, and noise. On higher-quality datasets like VR-NeRF and ScanNet++, where scenes feature diverse multi-scale visuals, all methods improve, yet ARTDECO still delivers the best performance. Outdoor evaluation covers large-scale free-motion captures (Fast-LIVO2) and forward-facing driving datasets (Waymo, KITTI, Matrix-City). ARTDECO consistently outperforms baselines, demonstrating robustness to scale variation. Qualitative comparisons (Fig. 4) show that ARTDECO, enabled by its multi-level Gaussian primitive design, captures fine details, large-scale structures, and high-fidelity geometry within a compact representation. Additional results are provided in Tabs. 8 - 28, Fig.5 in A.10.

**Tracking Results Analysis.** Tab. 2 summarizes the tracking performance on indoor and outdoor benchmarks. With loop closure and covariance-matrix filtering, ARTDECO achieves markedly higher localization accuracy than other 3DGS-based systems on challenging multi-scale indoor datasets (TUM, ScanNet++). On outdoor datasets such as Waymo, it also delivers competitive

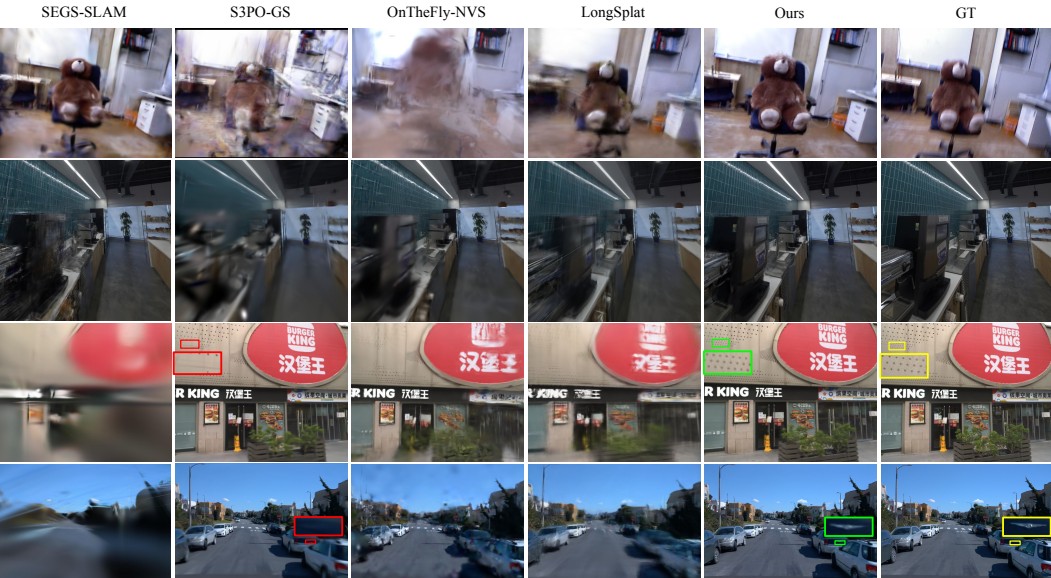

Figure 4: **Qualitative comparisons** against popular on-the-fly reconstruction baselines across diverse 3D scene datasets. ARTDECO consistently preserves high-quality rendering details in complex and diverse environments, particularly in the regions highlighted with colored rectangles.

Table 1: **Rendering comparisons against baselines** across indoor and outdoor datasets. We report visual quality metrics, average running time.

| Indoor-dataset Method | ScanNet++ PSNR↑ | SSIM↑ | LPIPS↓ | ScanNet PSNR↑ | SSIM↑ | LPIPS↓ | TUM PSNR↑ | SSIM↑ | LPIPS↓ | VR-NeRF PSNR↑ | SSIM↑ | LPIPS↓ | Training Time↓ |
|---|---|---|---|---|---|---|---|---|---|---|---|---|---|
| MonoGS | 16.71 | 0.682 | 0.600 | 18.87* | 0.780* | 0.629* | 17.78 | 0.602 | 0.573 | 13.88 | 0.560 | 0.420 | 14.08 min |
| S3PO-GS | 22.94 | 0.820 | 0.355 | 20.14 | 0.797 | 0.558 | 19.62 | 0.656 | 0.466 | 12.43 | 0.642 | 0.497 | 41.25 min |
| SEGS-SLAM | - | - | - | 19.73* | 0.839* | 0.365* | 19.69* | 0.743* | 0.307* | 31.62* | 0.896* | 0.232* | 10.84 min |
| OnTheFly-NVS | 18.01 | 0.761 | 0.386 | 15.36 | 0.708 | 0.494 | 19.72 | 0.719 | 0.380 | 27.30 | 0.872 | 0.310 | **2.29 min** |
| LongSplat | 24.94* | 0.827* | 0.260* | 19.27* | 0.754* | 0.404* | 25.09 | 0.804 | 0.272 | 25.74* | 0.832* | 0.321* | 442.96 min |
| Ours | **29.12** | **0.918** | **0.167** | **24.10** | **0.865** | **0.271** | **26.18** | **0.850** | **0.224** | **28.57** | **0.895** | **0.242** | 5.33 min |

| Outdoor-dataset Method | KITTI PSNR↑ | SSIM↑ | LPIPS↓ | Waymo PSNR↑ | SSIM↑ | LPIPS↓ | Fast-LIVO2 PSNR↑ | SSIM↑ | LPIPS↓ | MatrixCity PSNR↑ | SSIM↑ | LPIPS↓ | Training Time↓ |
|---|---|---|---|---|---|---|---|---|---|---|---|---|---|
| MonoGS | 14.56 | 0.489 | 0.767 | 19.34 | 0.752 | 0.627 | 18.87 | 0.598 | 0.699 | 19.36 | 0.593 | 0.736 | 16.52 min |
| S3PO-GS | 19.97 | 0.645 | 0.410 | 27.28 | 0.865 | 0.352 | 21.51 | 0.684 | 0.445 | 21.76 | 0.661 | 0.584 | 34.89 min |
| SEGS-SLAM | 14.03 | 0.463 | 0.488 | 19.01* | 0.698* | 0.502* | 24.58* | 0.773* | 0.307* | 25.57 | 0.784 | 0.366 | 8.75 min |
| OnTheFly-NVS | 16.89 | 0.579 | 0.471 | 25.53 | 0.820 | 0.360 | 18.76 | 0.618 | 0.497 | 21.36 | 0.687 | 0.451 | **0.74 min** |
| LongSplat | 16.86 | 0.532 | 0.447 | 25.61 | 0.795 | 0.326 | 26.37 | 0.792 | 0.276 | - | - | - | 313.60 min |
| Ours | **23.17** | **0.765** | **0.299** | **28.75** | **0.880** | **0.276** | **29.54** | **0.894** | **0.158** | 25.62 | 0.790 | 0.327 | 6.58 min |

*: majority of scenes successful; −: majority failed; Only compare fully successful methods.

performance. Further results on TUM (Second part of Tab. 2) demonstrate that ARTDECO consistently outperforms state-of-the-art non-3DGS SLAM methods, confirming its superior localization capability. Per-scene metrics, additional tracking results and qualitative trajectory comparisons are provided in Tabs. 30–34, Figs. 6–11 in A.10.

**Runtime Analysis.** We compare runtime across 3DGS-based methods on both indoor and outdoor datasets (Tab. 1). ARTDECO runs faster than all except OnTheFly-NVS, with its extra time cost primarily from pose estimation, a trade-off justified by the superior pose accuracy in Tab. 2.

Table 2: **Tracking comparisons.** For tracking evaluation, we compare against SLAM- and SFM-based 3D reconstruction methods on indoor and outdoor datasets, as well as state-of-the-art SLAM systems on the TUM dataset (Following MASt3R-SLAM, 9 scenes from TUM fr1). Our method consistently achieves lower ATE RMSE.

| Dataset | MonoGS | S3PO-GS | SEGS-SLAM | MASt3R-SLAM | OnTheFly-NVS | LongSplat | Ours |
|---|---|---|---|---|---|---|---|
| ScanNet++ | 1.217 | 0.632 | 0.245 | 0.025 | 0.891 | 0.602 | **0.018** |
| TUM | 0.244 | 0.117 | 0.073* | 0.031 | - | - | **0.025** |
| Waymo | 7.370 | 1.236 | - | - | 3.118 | 4.956 | **1.213** |

| Metric | ORB-SLAM3 | DPV-SLAM++ | DROID-SLAM | Go-SLAM | MASt3R-SLAM | Ours |
|---|---|---|---|---|---|---|
| ATE RMSE | - | 0.054 | 0.038 | 0.035 | 0.030 | **0.028** |

*: majority of scenes successful; –: majority failed; Only compare fully successful methods.

Table 3: **Quantitative results on ablation studies.** We separately listed the rendering metrics and ATE RMSE on ScanNet++ dataset for each ablation described in Sec. 4.3

| Front&Backend | Full | w/ SLAM (MASt3R $\to \pi^3$) | w/ Loop ($\pi^3 \to$ vggt) | w/o loop | w/ dense key frame |
|---|---|---|---|---|---|
| ATE RMSE | **0.018** | 0.374 | 0.096 | 0.057 | 0.094 |

| Mapper | Full | w/o level-of-detail | w/o implicit structure | w/o global feat | w/o mapper frame | w/o common frame |
|---|---|---|---|---|---|---|
| PSNR | **29.12** | 28.13 | 28.54 | 28.89 | 26.38 | 27.20 |
| SSIM | **0.918** | 0.912 | 0.914 | 0.916 | 0.898 | 0.904 |
| LPIPS | **0.167** | 0.180 | 0.175 | 0.170 | 0.229 | 0.211 |

## 4.3 Ablation Study

**Ablation on Localization.** We analyze the impact of backbone choice, loop closure, and frame categorization strategy on localization, as summarized in Tab. 3. We first ablate the feed-forward model used in the *frontend* and *backend* by replacing MASt3R (pairwise inference) with $\pi^3$ (multi-image inference). Although $\pi^3$ is trained on more diverse data, it lacks metric-scale capability and performs worse under varying viewpoints. In contrast, MASt3R better preserves consistent object proportions, resulting in more accurate pose estimation. Next, we ablate the loop-closure module by disabling it, which leads to a significant degradation in localization accuracy. Finally, we ablate the frame categorization strategy. Here, MF denotes mapper frames and KF denotes keyframes. Using both MFs and KFs (track w/ MF&KF) for inference provides additional temporal information, but unexpectedly reduces pose accuracy. This is because 3D foundation models often struggle with small-parallax inputs, producing ghosting and blur that corrupt point clouds and feature correspondences when the input sequence is overly dense.

**Ablation on Reconstruction.** We further ablate the effects of mapper frames, level-of-detail, and structural Gaussians on reconstruction, as shown in Tab. 3. MFs add richer multi-view constraints, while LoD and structural Gaussians yield more compact, regularized representations, together improving reconstruction fidelity and rendering quality.

## 5 Limitations

While ARTDECO achieves strong reconstruction and localization, it has several limitations. First, it partly depends on feed-forward 3D foundation models for correspondence and geometry, which, despite enabling fast and scalable inference, reduce robustness under noise, blur, or lighting changes, and suffer when inputs fall outside the training distribution. Second, the system assumes consistent illumination and sufficient parallax; violations such as low-texture surfaces, repetitive structures, or near-degenerate trajectories can cause drift or artifacts. These challenges suggest future work on incorporating uncertainty estimation, adaptive model selection, and stronger priors to improve generalization and reliability in real-world settings.

## 6 CONCLUSION

In this work, we present ARTDECO, a unified framework that advances on-the-fly 3D reconstruction from monocular image sequences. Beyond achieving strong results on standard indoor and outdoor benchmarks, ARTDECO demonstrates that feed-forward priors and structured Gaussian representations can be effectively combined within a single system to deliver both accuracy and efficiency. We see ARTDECO as a step toward practical large-scale deployment of real-to-sim pipelines, with promising applications in AR/VR, robotics, and digital twins.

## 7 ACKNOWLEDGMENTS

This work was funded in part by the National Key R&D Program of China (2022ZD0160201), Shanghai Artificial Intelligence Laboratory, the National Natural Science Foundation of China (Grant No. 62502247) and the HKU Startup Fund.

## ETHICS STATEMENT

This work focuses on SLAM (Simultaneous Localization and Mapping) algorithms and does not involve human subjects, private or sensitive data, or applications that pose direct societal risks. All experiments are conducted on publicly available datasets commonly used in the community. We believe that our research raises no concerns regarding human subjects, privacy, fairness, bias, or potential harmful use.

## REPRODUCIBILITY STATEMENT

We have taken steps to ensure the reproducibility of our results. All datasets used in our experiments are publicly available benchmarks, as detailed in Section 4.1 and the Appendix. We provide comprehensive descriptions of implementation details and experimental settings in these sections. Upon acceptance, we will release our source code and scripts to facilitate replication of all experiments. Together, these measures ensure that our results can be independently verified and reproduced.

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

THE USE OF LLM STATEMENT

We used GPT-5 as a general-purpose assistant during the preparation of this work. Specifically, GPT-5 was employed for (1) refining the writing style of the manuscript to improve clarity and readability, and (2) helping to organize and clean the code for presenting experimental results. GPT-5 was not involved in research ideation, algorithm design, or in producing any novel technical contributions. All ideas, methods, and experiments in this paper are solely the work of the authors.

## A  SUPPLEMENTARY MATERIAL

We organize the supplementary part as:

- **Sec. A.1** describes Implementation & evaluation protocol.
- **Sec. A.2** explains Jacobians and covariance transformation used in the *Frontend/Backend*.
- **Sec. A.3** discusses additional experimental details, such as $\pi^3$ ablations (multi-frame setup in frontend/backend and observations).
- **Sec. A.4** show an additional experiment with AnySplat Jiang et al. (2025b).
- **Sec. A.5** presents a comparison under low-texture and extreme motion scenarios to evaluate robustness.
- **Sec. A.6** presents more detailed ablation studies, including investigations into global_feat and Level of Detail (LOD).
- **Sec. A.7** presents a multi-resolution experiment to demonstrate the antialiasing capabilities of our method.
- **Sec. A.8** details the *Frontend* implementation (correspondences, residuals/weights, Gauss–Newton updates, and focal optimization when intrinsics are unknown).
- **Sec. A.9** details the *Backend* (loop-closure with ASMK and multi-frame 3D priors; global bundle adjustment).
- **Sec. A.10** provides extended experiments, per-scene metrics, and qualitative results.

### A.1  MORE IMPLEMENTATION DETAILS

**Comparison of Metrics.**  In previous methods, reconstruction metrics are often evaluated by removing the selected keyframes and evaluating the remaining frames. However, since each method selects different keyframes, the remaining frames vary, leading to slight differences in the images used to evaluate the Novel View Synthesis (NVS). For instance, selecting more keyframes results in fewer frames for evaluation, and vice versa. To ensure an absolutely fair comparison, we propose selecting one frame every eight frames for evaluation. The selected frames are not involved in the rendering process and are only used to optimize their poses.

We have modified the code for all our baselines to ensure that their evaluation metrics are computed using this approach, where every eighth frame is selected for evaluation, and the evaluated frames are not involved in the reconstruction supervision but are only used for pose optimization.

### A.2  JACOBIAN

**Jacobian w.r.t. Predicted Points.**  For completeness, we detail covariance transformation and related Jacobian in Sec. 3.1. Let $\pi(X, Y, Z) = \left(f_x X/Z + c_x, \ f_y Y/Z + c_y, \ \log Z\right)$ be the projection into image coordinates and log-depth with Jacobian $\mathbf{J}_\pi(\mathbf{x}) = \partial\pi/\partial\mathbf{x}$. The measurement-space covariance $\boldsymbol{\Sigma}_{ck} = \mathbf{J}_\pi(\mathbf{x}_{ck})\mathbf{R}_{kc}\,\boldsymbol{\Sigma}_c\,\mathbf{R}_{kc}^\top\mathbf{J}_\pi(\mathbf{x}_{ck})^\top$ can be derived, where $\mathbf{R}_{kc}$ is the rotation matrix of the relative pose $\mathbf{T}_{kc}$ between current frame and current keyframe. A residual is accepted only if $\det(\boldsymbol{\Sigma}_{ck}) < \tau$. Here, $\det(\boldsymbol{\Sigma}_{ck}) = \prod_i \lambda_i$ represents the generalized variance. The Jacobian is:

$$\mathbf{J}_\pi(\mathbf{x}_k) = \begin{bmatrix} \frac{f_x}{Z} & 0 & -\frac{f_x X}{Z^2} \\[6pt] 0 & \frac{f_y}{Z} & -\frac{f_y Y}{Z^2} \\[6pt] 0 & 0 & \frac{1}{Z} \end{bmatrix}_{(X,Y,Z)=\mathbf{x}_k} . \tag{4}$$

This criterion rejects measurements with excessive uncertainty while remaining rotation-invariant.

**Jacobian w.r.t. Focal.** In Sec. 3.1, besides the Jacobian with respect to the relative pose between the current frame and the keyframe, we also require the Jacobian of the projection with respect to the focal length $f$ when it is not provided.

Given a pixel $\mathbf{p}^c = (u^c, v^c)$ in the current frame, we first recover its camera-coordinate point $\mathbf{P}^c = (X^c, Y^c, Z^c)^\top$ using the depth $Z^c$ predicted by MASt3R. This point can be re-projected into the current frame as

$$\mathbf{P}'^c = \begin{bmatrix} X'^c \\ Y'^c \\ Z'^c \end{bmatrix} = (Z^c) \cdot \begin{bmatrix} (u^c - c_x)/f \\ (v^c - c_y)/f \\ 1 \end{bmatrix},$$

where $(f, c_x, c_y)$ denote the intrinsic parameters with $f_x = f_y = f$.

Given the relative pose $T_{kc}$ between the current frame and a keyframe $k$, the transformed point is

$$\mathbf{P}^k = T_{kc}\,\mathbf{P}'^c = \begin{bmatrix} X^k \\ Y^k \\ Z^k \end{bmatrix}.$$

Finally, projecting $\mathbf{P}^k$ back to the image plane yields

$$\pi(\mathbf{P}^k) = \begin{bmatrix} u \\ v \\ \log Z \end{bmatrix} = \begin{bmatrix} f\frac{X^k}{Z^k} + c_x \\ f\frac{Y^k}{Z^k} + c_y \\ \log Z^k \end{bmatrix}.$$

**Direct Effect of $f$:** The direct effect refers to how $f$ directly influences the projection of $(X^k, Y^k, Z^k)$ onto the image plane. This Jacobian term is straightforward and is given by the following matrix:

$$\frac{\partial(u, v, \log Z)}{\partial f} = \begin{bmatrix} \frac{X^k}{Z^k} \\ \frac{Y^k}{Z^k} \\ 0 \end{bmatrix}.$$

**Indirect Effect of $f$:** The indirect effect arises from how the depth-dependent point $\mathbf{P}'^c$ (which depends on $f$) affects the final transformed point $\mathbf{P}^k$. This term is computed by differentiating the reconstruction of $\mathbf{P}'^c$ with respect to $f$:

$$\frac{\partial(X^k, Y^k, Z^k)}{\partial f} = T_{kc}\frac{\partial\mathbf{P}'^c}{\partial f}, \quad \frac{\partial\mathbf{P}'^c}{\partial f} = (Z^c) \cdot \begin{bmatrix} -\frac{(u^c - c_x)}{f^2} \\ -\frac{(v^c - c_y)}{f^2} \\ 0 \end{bmatrix}.$$

**Full Jacobian:** Finally, combining the direct and indirect effects, the total Jacobian $\mathbf{J}_{\pi,f}$ is the sum of these two parts:

$$\mathbf{J}_{\pi,f}(\mathbf{P}^k) = \frac{\partial(u, v, \log Z)}{\partial f} + \frac{\partial(u, v, \log Z)}{\partial\mathbf{P}^k} \cdot \frac{\partial\mathbf{P}^k}{\partial f} = \begin{bmatrix} \frac{X^k}{Z^k} \\ \frac{Y^k}{Z^k} \\ 0 \end{bmatrix} + \begin{bmatrix} \frac{f}{Z^k} & 0 & -\frac{fX^k}{(Z^k)^2} \\ 0 & \frac{f}{Z^k} & -\frac{fY^k}{(Z^k)^2} \\ 0 & 0 & \frac{1}{Z^k} \end{bmatrix} \begin{bmatrix} -\frac{(u^c - c_x)}{f^2} \\ -\frac{(v^c - c_y)}{f^2} \\ 0 \end{bmatrix}.$$

### A.3 ADDITIONAL EXPERIMENTS

In Sec. 4.3, we replaced the MASt3R model, which is designed for two-frame inference, with the $\pi^3$ model, which supports multi-frame inference. However, the ablation experiments show that the $\pi^3$ model struggles to maintain consistent size ratios between the same objects across different viewpoints and disparities. In this section, we provide a detailed description of the experimental setup and results for replacing MASt3R with the $\pi^3$ model.

Table 4: Quantitative comparison between AnySplat Jiang et al. (2025b) and our method. We report Tracking Error (lower is better), PSNR, SSIM, and LPIPS across different view counts.

| Method | ScanNet++ (In-domain) | | | | Fast-LiVO2 (Out-domain) | | | |
|---|---|---|---|---|---|---|---|---|
| | Tracking ↓ | PSNR ↑ | SSIM ↑ | LPIPS ↓ | Tracking ↓ | PSNR ↑ | SSIM ↑ | LPIPS ↓ |
| *32 views* | | | | | | | | |
| AnySplat | 0.082 | 23.41 | 0.822 | **0.176** | 0.235 | 24.21 | 0.772 | 0.198 |
| Ours | **0.034** | **26.53** | **0.856** | 0.201 | **0.065** | **27.96** | **0.853** | **0.187** |
| *100 views* | | | | | | | | |
| AnySplat | 0.128 | 21.37 | 0.780 | 0.227 | 0.426 | 21.28 | 0.672 | 0.287 |
| Ours | **0.076** | **24.36** | **0.841** | **0.218** | **0.270** | **24.56** | **0.788** | **0.255** |

**Frontend with $\pi^3$ Inference and Keyframe Selection.** In the frontend, we configure the $\pi^3$ model to infer $k$ images at once. As images are streamed, the frontend waits until $k$ images are accumulated before performing a single inference. After inference, the Ray optimization method is applied to compute the correspondence between these frames, followed by a local bundle adjustment (LBA) based on Sim(3) pose optimization among the $k$ frames. Upon completing this process, we follow the keyframe selection strategy described in Sec. 3.1, and the selected keyframes are then passed to the backend.

After the first inference, for each subsequent inference, the frontend first selects the $l$ most recent keyframes and waits for the next $k - l$ frames to complete the batch of $k$ frames. Then, another inference and LBA are performed, and the keyframe selection is updated accordingly.

**Backend with $\pi^3$ Inference and Global Bundle Adjustment.** The backend receives the selected keyframes from the frontend. Similar to the frontend strategy, the backend waits until $k$ keyframes are accumulated before performing $\pi^3$ inference. Once enough keyframes are available, correspondence is computed between them, followed by a Global Bundle Adjustment (GBA) to optimize the poses.

In each inference step, the backend also includes $l$ historical keyframes into the inference window, ensuring sufficient overlap between keyframes from multiple inference steps. Unlike the frontend, however, the backend leverages the streamable ASMK algorithm from the MASt3R-SLAM framework to compute the most similar historical keyframes to the current keyframe. These selected keyframes are then included in the inference window to perform the optimization.

**Mapper with Keyframe Reception.** We did not modify the mapper code. The mapper simply receives the keyframes sent by the backend and proceeds with the reconstruction following the process described in Sec. 3.3.

## A.4 COMPARISON WITH FEED-FORWARD METHODS

As summarized in Table 4, we evaluate our system against state-of-the-art feed-forward approaches, specifically AnySplat. While feed-forward methods offer a simplified pipeline, they suffer from several inherent bottlenecks: restricted operating resolutions, prohibitive memory growth on extended sequences, and the absence of a backend optimization module to rectify accumulated pose drift.

Our quantitative results on ScanNet++ Yeshwanth et al. (2023) (in-domain) and Fast-LiVO2 Zheng et al. (2024) (out-domain) highlight these deficiencies. Notably, as the sequence length increases from 32 to 100 views, AnySplat exhibits a significant performance degradation; for instance, its tracking error on ScanNet++ rises from 0.082 to 0.128, and PSNR drops by nearly 2 dB. In contrast, our modular SLAM-style design maintains superior stability and reconstruction fidelity. The consistent gains across both datasets, especially the robust performance on the out-domain Fast-LiVO2, underscore the necessity of systematic post-optimization for reliable long-sequence SLAM and high-quality radiance field synthesis.

Table 5: Quantitative evaluation of tracking accuracy (ATE RMSE [m] ↓) and reconstruction quality (PSNR ↑, SSIM ↑, LPIPS ↓). The symbol "/" indicates a system crash, tracking failure, or the lack of rendering capability for that specific baseline.

| Method | TUM (Low Texture) | | | | TartanAirV2 (Extreme Motion) | | | |
|---|---|---|---|---|---|---|---|---|
| | ATE ↓ | PSNR ↑ | SSIM ↑ | LPIPS ↓ | ATE ↓ | PSNR ↑ | SSIM ↑ | LPIPS ↓ |
| LDSO | 0.207 | / | / | / | 0.230 | / | / | / |
| ORB-SLAM3 | / | / | / | / | / | / | / | / |
| Go-SLAM | 0.371 | / | / | / | 0.258 | / | / | / |
| DROID-SLAM | 0.407 | / | / | / | 0.301 | / | / | / |
| DPV-SLAM++ | 0.054 | / | / | / | 0.223 | / | / | / |
| MASt3R-SLAM | 0.076 | / | / | / | 0.537 | / | / | / |
| NICER-SLAM | / | / | / | / | 1.666 | 12.43 | 0.445 | 0.671 |
| MonoGS | 0.671 | 27.43 | 0.912 | 0.353 | / | / | / | / |
| S3PO-GS | 0.524 | 28.97 | 0.916 | 0.340 | 1.260 | 14.87 | 0.598 | 0.631 |
| SEGS-SLAM | / | / | / | / | / | / | / | / |
| OnTheFly-NVS | 0.565 | 26.62 | 0.918 | 0.355 | 1.116 | 15.34 | 0.635 | 0.527 |
| LongSplat | 0.231 | 31.68 | 0.911 | 0.296 | 1.244 | 17.26 | 0.639 | 0.500 |
| Ours | **0.072** | **33.74** | **0.950** | **0.264** | **0.095** | **23.17** | **0.813** | **0.268** |

## A.5 ROBUSTNESS EVALUATION

To verify the robustness and versatility of our system, we conduct extensive evaluations against a broad spectrum of baselines. These include classic SLAM frameworks such as ORB-SLAM3 Campos et al. (2021), DROID-SLAM Teed & Deng (2021), Go-SLAM Zhang et al. (2023), DPV-SLAM++ Lipson et al. (2024), LDSO Gao et al. (2018), and the recent MASt3R-SLAM Murai et al. (2025). We further compare against state-of-the-art 3DGS-based SLAM methods, including MonoGS Feng et al. (2025), S3PO-GS Cheng et al. (2025b), and SEGS-SLAM Wen et al. (2025), as well as advanced reconstruction approaches like OnTheFly-NVS Meuleman et al. (2025), NICER-SLAM Zhu et al. (2022), LongSplat Lin et al. (2025).

Our evaluation focuses on two challenging scenarios: the TUM RGB-D Structure vs. Texture category to assess performance in low-texture environments, and the TartanAirV2 Hard sequences to test tracking and mapping stability under extreme camera motion. As shown in Table 5, many existing methods suffer from tracking failures or system crashes (denoted by /) in these demanding conditions. In contrast, our method achieves the lowest ATE RMSE on both datasets, demonstrating superior localization stability.

Furthermore, the reconstruction quality results in Table 5 highlight our system's ability to maintain high-fidelity rendering even when geometric cues are sparse or motion blur is prevalent. Our approach significantly outperforms both traditional and learning-based baselines across all metrics (PSNR, SSIM, and LPIPS), establishing its effectiveness for robust spatial AI applications.

## A.6 ADDITIONAL ABLATION STUDIES

To further validate the contribution of each core component in our system, we provide a comprehensive set of ablation experiments. These studies focus on the impact of mapper frames, our Level-of-Detail (LoD) strategy, and the structural design of Gaussians.

**Impact of Mapper Frames.** Removing mapper frames forces the Gaussian initialization process to rely solely on keyframes, which reduces the number of initialization frames by 50–70%. As shown in Table 6, this leads to overly sparse geometry and a significant drop in reconstruction quality (2.74 dB PSNR), confirming that mapper frames are essential for maintaining density and stability in online mapping.

**Level-of-Detail (LoD) and Gaussian Selection.** Unlike prior methods like Octree-GS that assume a static, complete point cloud, our system is designed for incremental growth. By introducing a pixel-wise, point-cloud-aware selection attribute, we achieve a significant reduction in the number of

Table 6: Consolidated ablation results. #Render denotes the number of rendered Gaussians in thousands (K), and MEM denotes the GPU memory usage in MB. The best results are highlighted in **bold**.

| Method | PSNR ↑ | SSIM ↑ | LPIPS ↓ | #Render (K) ↓ | MEM (MB) ↓ |
|---|---|---|---|---|---|
| **Ours (Full)** | **29.12** | **0.918** | **0.167** | **130** | 127.2 |
| *(1) Mapping Strategy* | | | | | |
| Ours w/o mapper frames | 26.38 | 0.898 | 0.229 | 73 | **78.8** |
| *(2) LoD & Selection Mechanism* | | | | | |
| Ours w/o gs selection | 29.12 | 0.918 | 0.167 | 176 | 127.2 |
| Ours w/o level-of-detail | 28.13 | 0.912 | 0.180 | 181 | 133.3 |
| *(3) Structural Feature Ablation (w/o LoD)* | | | | | |
| w/o global_feat | 27.95 | 0.910 | 0.197 | 184 | 161.5 |
| global_feat (8) + local_feat (24) | 28.11 | 0.912 | 0.181 | 184 | 141.2 |
| global_feat (16) + local_feat (16) | 28.13 | 0.912 | 0.180 | 181 | 133.3 |
| global_feat (24) + local_feat (8) | 28.03 | 0.911 | 0.185 | 182 | 127.6 |
| w/o local_feat | 27.98 | 0.910 | 0.191 | 183 | 121.7 |

rendered Gaussians (from 176K to 130K) without sacrificing fidelity. Without the LoD mechanism, the system suffers from increased computational overhead and degraded high-frequency details.

**Structural Gaussian Design.** We analyze the effectiveness of decoding Gaussian attributes from a combination of global features and local features. Our experiments indicate that a balanced feature distribution (e.g., 16-dimensional global and 16-dimensional local features) provides the best trade-off between reconstruction accuracy (PSNR/LPIPS) and memory efficiency. This structured approach enforces local coherence and accelerates convergence compared to unstructured baselines.

## A.7 ANTI-ALIASING EXPERIMENT

To evaluate the efficacy of our anti-aliasing selection strategy, we conduct a multi-resolution ablation study on the "Bicycle" scene from the Mip-NeRF 360 dataset Barron et al. (2022), adhering to the experimental configuration established by Octree-GS Ren et al. (2025). Our method introduces a scale-aware filtering mechanism based on a modified distance metric $d_{rs} = s \cdot d_r$, where $s$ denotes the scale factor determined by the ratio of the standard resolution to the current rendering resolution. During the rendering process, a Gaussian primitive is included if $d_{rs} \leq d_{max}$ and excluded if $d_{rs} \geq 2 \cdot d_{max}$. As summarized in Table 1 and illustrated in Figure 10 of the supplementary material, this strategy significantly enhances rendering efficiency by drastically reducing the Gaussian count at lower resolutions. For instance, at $1/8$ resolution, the number of rendered primitives is pruned from $1131K$ to a mere $29K$ while maintaining comparable PSNR and SSIM metrics. Furthermore, the anti-aliasing module effectively preserves structural integrity across varying scales; without this mechanism, the bicycle frame appears disproportionately thick at low resolutions, while fine structures like spokes become excessively thin at high resolutions. By selectively rendering only scale-appropriate Gaussians, our approach ensures consistent geometric proportions and high-fidelity reconstruction across all resolution levels.

Table 7: Multi-resolution Ablation Study on the Mip-NeRF 360 Barron et al. (2022) "Bicycle" Scene. The results demonstrate that our anti-aliasing strategy significantly reduces the number of rendered Gaussians (#Render) at lower scales while maintaining robust reconstruction quality across four resolution levels ($1\times$ to $1/8\times$).

| Scale | 1× | | | 1/2× | | | 1/4× | | | 1/8× | | |
|---|---|---|---|---|---|---|---|---|---|---|---|---|
| Method | PSNR ↑ | SSIM ↑ | #Render | PSNR ↑ | SSIM ↑ | #Render | PSNR ↑ | SSIM ↑ | #Render | PSNR ↑ | SSIM ↑ | #Render |
| Ours w/o anti-alias | 20.27 | 0.397 | 1083K | 21.67 | 0.475 | 1085K | 23.60 | 0.614 | 1097K | 25.01 | 0.743 | 1131K |
| **Ours** | **20.48** | **0.417** | **1008K** | **21.83** | **0.491** | **657K** | **23.62** | **0.621** | **167K** | 24.61 | 0.732 | **29K** |

A.8 Details in *Frontend*

For completeness, we also provide the details for pose/focal optimization in the frontend. Inspired by MASt3R-SLAM Murai et al. (2025), we use MASt3R Leroy et al. (2024) to estimate dense pixel-wise correspondences and associated 3D points between the current frame and the current keyframe. We denote the set as $\mathcal{C} = \{(\mathbf{p}_m^c, \mathbf{p}_m^k, \mathbf{P}_m^c, \mathbf{P}_m^k)\}_{m=1}^M$, where $\mathbf{p}_m^c, \mathbf{p}_m^k \in \mathbb{R}^2$ are pixel coordinates and $\mathbf{P}_m^c = [X_m^c, Y_m^c, Z_m^c] \in \mathbb{R}^3, \mathbf{P}_m^k = [X_m^k, Y_m^k, Z_m^k] \in \mathbb{R}^3$ are MASt3R-inferred 3D points expressed in the current-frame and keyframe coordinate systems, respectively. Given $\mathcal{C}$, we project the current-frame 3D points into the keyframe image plane and form reprojection residuals, which are minimized using a Gauss–Newton procedure. We estimate the similarity transform $\mathbf{T}_{kc} = (s_{kc}, \mathbf{R}_{kc}, \mathbf{t}_{kc}) \in \mathrm{Sim}(3)$ with $s > 0$, $\mathbf{R}_{kc} \in \mathrm{SO}(3)$, $\mathbf{t}_{kc} \in \mathbb{R}^3$; when intrinsics are unknown, we additionally optimize a single focal length $f$.

For each correspondence, we compose the similarity transform and projection directly into the per-point residual:

$$\mathbf{r}_m = \begin{bmatrix} \mathbf{p}_m^k - \hat{\mathbf{p}}_m^k \\ \log(Z_m^k) - \log(\hat{Z}_m^k) \end{bmatrix}, \qquad \hat{\mathbf{P}}_m^k = s_{kc}\,\mathbf{R}\,\mathbf{P}_m^c + \mathbf{t}, \hat{\mathbf{p}}_m^k = \pi_{\mathbf{K}(f)}(\hat{\mathbf{P}}_m^k), \tag{5}$$

where $\mathbf{K}(f)$ is the camera intrinsic and $\pi_{\mathbf{K}(f)}$ is the pinhole projection function.

To encode measurement confidence and balance pixel vs. depth scales, each correspondence $m$ is assigned a positive semidefinite weight matrix $\mathbf{W}_m \in \mathbb{R}^{3\times3}$. The (Mahalanobis) squared residual is

$$t_m = \mathbf{r}_m^\top \mathbf{W}_m \mathbf{r}_m \geq 0, \qquad s_m = \sqrt{t_m} \text{ is the residual norm.} \tag{6}$$

We use the robust Huber kernel to eliminate the influence of outliers.

$$\omega_m = \begin{cases} 1, & s_m \leq \delta, \\ \dfrac{\delta}{s_m + \varepsilon}, & s_m > \delta, \end{cases} \qquad \widetilde{\mathbf{W}}_m = \omega_m\,\mathbf{W}_m, \tag{7}$$

with a small $\varepsilon > 0$ for numerical stability.

With robust weights, the objective becomes a weighted least-squares problem:

$$E_{rob} = \frac{1}{2}\sum_{i=1}^M \mathbf{r}_m^\top \widetilde{\mathbf{W}}_m \mathbf{r}_m \tag{8}$$

Linearizing $\mathbf{r}_m$ at the current estimate yields the normal equations

$$\left(\sum_{i=1}^M \mathbf{J}_m^\top \widetilde{\mathbf{W}}_m \mathbf{J}_m\right)\Delta\boldsymbol{\theta} = \sum_{i=1}^M \mathbf{J}_m^\top \widetilde{\mathbf{W}}_m \mathbf{r}_m, \qquad \mathbf{J}_m = \frac{\partial \mathbf{r}_m}{\partial \boldsymbol{\theta}} \in \mathbb{R}^{3\times d}, \tag{9}$$

where $d = 7$ for pose-only $\boldsymbol{\theta} = \{\mathbf{T}_{kc}\}$ and $d = 8$ if the focal $f$ is also optimized. We update the similarity transform by the $\mathrm{Sim}(3)$ exponential map $\mathbf{T}_{kc} \leftarrow \exp_{\mathrm{Sim}(3)}(\Delta\boldsymbol{\xi}_{\mathrm{sim}})\,\mathbf{T}_{kc}$, and, when applicable, the focal by $f \leftarrow f + \Delta f$. Here $\Delta\boldsymbol{\xi}_{\mathrm{sim}} \in \mathbb{R}^7$ is the minimal $\mathrm{Sim}(3)$ increment and $\Delta\boldsymbol{\theta} = [\Delta\boldsymbol{\xi}_{\mathrm{sim}}^\top, \Delta f]^\top$ if $f$ is included.

A.9 Details in *Backend*

**Details about Loop Closure.** For the current keyframe received from the backend, we first compute the similarity score between the current keyframe and all historical keyframes using the ASMK algorithm. Only historical keyframes with a similarity score greater than 0.005 are considered as valid matches. However, the ASMK-based loop closure detection algorithm, relying solely on 2D features, lacks 3D prior knowledge and is not well-suited as a robust loop closure metric. Recently, multi-frame 3D foundation models, such as VGGT, $\pi^3$, have provided a new approach for loop closure detection by incorporating rich 3D priors.

Based on this idea, we first calculate the maximum time gap between the current keyframe and the valid historical keyframes. If the maximum time gap exceeds 10, we assume that the current keyframe has a high probability of being similar to the historical keyframes. In such cases, we revisit the previously computed ASMK score and select the top $N_a$ most similar keyframes. These keyframes are then used for $\pi^3$ inference alongside the current keyframe.

As long as there is overlap between the keyframes (even not), 3D foundation models can leverage the rich 3D priors to infer point clouds in the same coordinate system. We then calculate the angular error between corresponding points in different frames. If the angular error is below a certain threshold and the distance between the points is also small, we consider them as a match. After calculating the ratio of matched pixels between the current frame and historical keyframes to the total number of pixels, we sort the matches by their ratio. We identify the top three frames where the ratio exceeds 0.15 and consider these as loop closures.

In an ideal scenario, every incoming keyframe would be inferred with all historical keyframes using $\pi^3$ inference. While this approach would significantly enhance the loop closure ability, it is computationally expensive and time-consuming. Therefore, we opted for the aforementioned approach based on ASMK pre-filtering as a compromise. This method effectively balances both accuracy and speed, as demonstrated by our experiments, making it more practical for real-time applications.

**Details about Global Bundle Adjustment.** For completeness, we also provide the details for global bundle adjustment. For a correspondence $m$ between frames $i$ and $j$, let $\mathbf{P}_m^i \in \mathbb{R}^3$ be the 3D point in frame-$i$ and $\mathbf{p}_m^j \in \mathbb{R}^2$ the observed pixel in frame-$j$ with log-depth $\ell_m^j$. The world poses are $\mathbf{T}_{wi} = (s_i, \mathbf{R}_i, \mathbf{t}_i) \in \mathrm{Sim}(3)$ and $\mathbf{T}_{wj} = (s_j, \mathbf{R}_j, \mathbf{t}_j) \in \mathrm{Sim}(3)$. The world point and its coordinates in frame-$j$ are

$$\hat{\mathbf{P}}_m^w = s_i \mathbf{R}_i \mathbf{P}_m^i + \mathbf{t}_i, \qquad \hat{\mathbf{P}}_m^j = s_j^{-1} \mathbf{R}_j^\top (\hat{\mathbf{P}}_m^w - \mathbf{t}_j),$$

with $(\hat{X}_m^j, \hat{Y}_m^j, \hat{Z}_m^j)^\top = \hat{\mathbf{P}}_m^j$. Using intrinsics $\mathbf{K}$, the projection is

$$\hat{\mathbf{p}}_m^j = \pi_{\mathbf{K}}(\hat{\mathbf{P}}_m^j) = \left[ f_x \hat{X}_m^j / \hat{Z}_m^j + c_x, \ f_y \hat{Y}_m^j / \hat{Z}_m^j + c_y \right]^\top,$$

and we stack the image-plane and log-depth terms into

$$\mathbf{r}_m = \begin{bmatrix} \hat{\mathbf{p}}_m^j - \mathbf{p}_m^j \\ \log \hat{Z}_m^j - \ell_m^j \end{bmatrix} \in \mathbb{R}^3. \tag{10}$$

The Jacobian of $\mathbf{r}_m$ with respect to $\hat{\mathbf{P}}_m^j$ is

$$\frac{\partial \mathbf{r}_m}{\partial \hat{\mathbf{P}}_m^j} = \begin{bmatrix} \frac{f_x}{\hat{Z}_m^j} & 0 & -\frac{f_x \hat{X}_m^j}{(\hat{Z}_m^j)^2} \\ 0 & \frac{f_y}{\hat{Z}_m^j} & -\frac{f_y \hat{Y}_m^j}{(\hat{Z}_m^j)^2} \\ 0 & 0 & \frac{1}{\hat{Z}_m^j} \end{bmatrix}. \tag{11}$$

To optimize the poses, we apply left-multiplicative $\mathrm{Sim}(3)$ perturbations $\delta\boldsymbol{\xi}_i, \delta\boldsymbol{\xi}_j \in \mathbb{R}^7$ to $\mathbf{T}_{wi}$ and $\mathbf{T}_{wj}$, respectively. With the skew operator $[\cdot]_\times$ and $\hat{\mathbf{P}}_m^w$ defined above, the induced differentials of the point in frame-$j$ are

$$\frac{\partial \hat{\mathbf{P}}_m^j}{\partial \delta\boldsymbol{\xi}_i} = s_j^{-1} \mathbf{R}_j^\top \begin{bmatrix} \mathbf{I}_3 & -[\hat{\mathbf{P}}_m^w]_\times & \hat{\mathbf{P}}_m^w \end{bmatrix} \in \mathbb{R}^{3\times 7},$$

$$\frac{\partial \hat{\mathbf{P}}_m^j}{\partial \delta\boldsymbol{\xi}_j} = \begin{bmatrix} -\mathbf{I}_3 & [\hat{\mathbf{P}}_m^j]_\times & -\hat{\mathbf{P}}_m^j \end{bmatrix} \in \mathbb{R}^{3\times 7}, \tag{12}$$

which combine with the projection Jacobian via the chain rule:

$$\frac{\partial \mathbf{r}_m}{\partial \delta\boldsymbol{\xi}_i} = \frac{\partial \mathbf{r}_m}{\partial \hat{\mathbf{P}}_m^j} \frac{\partial \hat{\mathbf{P}}_m^j}{\partial \delta\boldsymbol{\xi}_i}, \qquad \frac{\partial \mathbf{r}_m}{\partial \delta\boldsymbol{\xi}_j} = \frac{\partial \mathbf{r}_m}{\partial \hat{\mathbf{P}}_m^j} \frac{\partial \hat{\mathbf{P}}_m^j}{\partial \delta\boldsymbol{\xi}_j}. \tag{13}$$

Finally, to initialize reliable 3D supervision for mapping after the pose update, for the current keyframe $k$ with points $\{\mathbf{X}_n^k\}_{n=1}^{N_k}$ and related keyframes $\mathcal{N}_k$, we project each point to every $j \in \mathcal{N}_k$,

compute the mean reprojection error $e_n$, assign a confidence $c_n$ by thresholding, and pass the pair to the mapping thread:

$$e_n = \frac{1}{|\mathcal{N}_k|} \sum_{j \in \mathcal{N}_k} \left\| \mathbf{u}_n^j - \pi_{\mathbf{K}}(\mathbf{T}_{jk} \mathbf{X}_n^k) \right\|_2,$$

$$c_n = \begin{cases} 1, & e_n \leq \tau_{\mathrm{proj}}, \\ \dfrac{1}{e_n - \tau_{\mathrm{proj}} + 1}, & e_n > \tau_{\mathrm{proj}}, \end{cases} \quad (14)$$

yielding the package $\{(\mathbf{X}_n^k, c_n)\}_{n=1}^{N_k}$ for Gaussian-primitive supervision in the mapping thread.

### A.10 MORE EXPERIMENTS

In this section, we provide the specific per-scene metrics for each dataset used to support the average metrics reported in the main text through Tab. 8- 34. These include reconstruction metrics such as PSNR, SSIM, and LPIPS for the following datasets: KITTI, Waymo, MatrixCity, Fast-LIVO2, Scan-Net, ScanNet++, VR-NeRF, and TUM. We also provide qualitative results related to reconstruction and tracking trajectories, as shown in Fig 6 7 11.

Table 8: PSNR on the Fast-LIVO2 Dataset

| Method | CBD_Building_01 | HKU_Campus | Red_Sculpture | Retail_Street | SYSU |
|---|---|---|---|---|---|
| MonoGS | 19.86 | 21.70 | 15.50 | 18.05 | 19.23 |
| SEGS-SLAM | 26.26 | _29.55_ | - | 18.49 | 24.01 |
| S3PO-GS | 17.47 | 25.47 | 18.89 | _24.30_ | 21.42 |
| OnTheFly-NVS | 17.79 | 21.46 | 17.50 | 17.67 | 19.39 |
| LongSplat | _29.25_ | 29.45 | _24.97_ | 23.10 | _25.07_ |
| Ours | **31.11** | **30.89** | **26.29** | **29.20** | **30.22** |

Table 9: SSIM on the Fast-LIVO2 Dataset

| Method | CBD_Building_01 | HKU_Campus | Red_Sculpture | Retail_Street | SYSU |
|---|---|---|---|---|---|
| MonoGS | 0.698 | 0.608 | 0.518 | 0.554 | 0.610 |
| SEGS-SLAM | 0.880 | _0.847_ | - | 0.589 | _0.777_ |
| S3PO-GS | 0.645 | 0.705 | 0.620 | _0.782_ | 0.669 |
| OnTheFly-NVS | 0.677 | 0.628 | 0.590 | 0.558 | 0.635 |
| LongSplat | _0.891_ | 0.832 | _0.786_ | 0.708 | 0.742 |
| Ours | **0.940** | **0.871** | **0.861** | **0.901** | **0.899** |

Table 10: LPIPS on the Fast-LIVO2 Dataset

| Method | CBD_Building_01 | HKU_Campus | Red_Sculpture | Retail_Street | SYSU |
|---|---|---|---|---|---|
| MonoGS | 0.623 | 0.687 | 0.778 | 0.750 | 0.658 |
| SEGS-SLAM | 0.213 | _0.205_ | - | 0.516 | _0.292_ |
| S3PO-GS | 0.592 | 0.448 | 0.505 | _0.232_ | 0.451 |
| OnTheFly-NVS | 0.490 | 0.470 | 0.500 | 0.528 | 0.498 |
| LongSplat | _0.179_ | 0.258 | _0.304_ | 0.315 | 0.322 |
| Ours | **0.108** | **0.199** | **0.205** | **0.127** | **0.151** |

Table 11: PSNR on the TUM Dataset

| Method | f1_360 | f1_desk | f1_desk2 | f1_floor | f1_plant | f1_room | f1_rpy | f1_teddy | f1_xyz | f2_xyz | f3_office |
|---|---|---|---|---|---|---|---|---|---|---|---|
| MonoGS | 16.17 | 14.86 | 14.96 | 20.71 | 17.46 | 15.38 | 16.28 | 16.50 | 21.89 | 21.23 | 20.10 |
| SEGS-SLAM | 19.43 | 19.81 | 18.44 | 21.75 | 17.33 | - | 18.44 | 15.37 | 20.68 | 19.54 | 26.14 |
| S3PO-GS | 16.70 | 20.09 | 18.52 | 22.69 | 18.41 | 17.14 | 16.67 | 19.02 | 22.78 | 23.06 | 20.74 |
| OnTheFly-NVS | 18.44 | 18.91 | 18.38 | 25.43 | 15.84 | 17.26 | 20.86 | 16.27 | 25.69 | 20.04 | 19.80 |
| LongSplat | **27.07** | 25.35 | 25.48 | 28.14 | 21.27 | 23.52 | 22.81 | 22.36 | **26.75** | 27.46 | 25.80 |
| Ours | 26.19 | **26.04** | **25.54** | **29.43** | **24.06** | **25.23** | **24.92** | **23.30** | 26.50 | **29.91** | **26.92** |

Table 12: SSIM on the TUM Dataset

| Method | f1_360 | f1_desk | f1_desk2 | f1_floor | f1_plant | f1_room | f1_rpy | f1_teddy | f1_xyz | f2_xyz | f3_office |
|---|---|---|---|---|---|---|---|---|---|---|---|
| MonoGS | 0.583 | 0.529 | 0.552 | 0.586 | 0.581 | 0.542 | 0.575 | 0.539 | 0.738 | 0.698 | 0.698 |
| SEGS-SLAM | 0.751 | 0.775 | 0.720 | 0.752 | 0.641 | - | 0.718 | 0.622 | 0.821 | 0.769 | 0.861 |
| S3PO-GS | 0.602 | 0.680 | 0.650 | 0.625 | 0.616 | 0.597 | 0.596 | 0.614 | 0.762 | 0.752 | 0.723 |
| OnTheFly-NVS | 0.725 | 0.712 | 0.713 | 0.783 | 0.600 | 0.662 | 0.755 | 0.594 | 0.870 | 0.730 | 0.760 |
| LongSplat | 0.826 | 0.833 | 0.842 | 0.767 | 0.695 | 0.790 | 0.783 | 0.736 | **0.888** | 0.879 | 0.799 |
| Ours | **0.850** | **0.861** | **0.859** | **0.838** | **0.797** | **0.838** | **0.847** | **0.768** | 0.883 | **0.922** | **0.882** |

Table 13: LPIPS on the TUM Dataset

| Method | f1_360 | f1_desk | f1_desk2 | f1_floor | f1_plant | f1_room | f1_rpy | f1_teddy | f1_xyz | f2_xyz | f3_office |
|---|---|---|---|---|---|---|---|---|---|---|---|
| MonoGS | 0.642 | 0.664 | 0.661 | 0.736 | 0.572 | 0.679 | 0.509 | 0.654 | 0.326 | 0.355 | 0.511 |
| SEGS-SLAM | 0.361 | 0.244 | 0.358 | 0.277 | 0.399 | - | 0.336 | 0.415 | 0.205 | 0.270 | 0.200 |
| S3PO-GS | 0.551 | 0.433 | 0.506 | 0.634 | 0.461 | 0.571 | 0.507 | 0.482 | 0.296 | 0.270 | 0.420 |
| OnTheFly-NVS | 0.445 | 0.396 | 0.404 | 0.305 | 0.501 | 0.440 | 0.353 | 0.505 | 0.199 | 0.326 | 0.307 |
| LongSplat | 0.324 | 0.266 | 0.270 | 0.305 | 0.406 | 0.255 | 0.267 | 0.300 | **0.127** | 0.152 | 0.325 |
| Ours | **0.279** | **0.220** | **0.238** | **0.233** | **0.263** | **0.251** | **0.235** | **0.298** | 0.174 | **0.080** | **0.191** |

Table 14: PSNR on the ScanNet Dataset

| Method | scene0000_00 | scene0059_00 | scene0106_00 | scene0169_00 | scene0181_00 | scene0207_00 |
|---|---|---|---|---|---|---|
| MonoGS | - | 18.09 | 18.03 | 19.71 | 19.37 | 19.16 |
| SEGS-SLAM | - | - | 17.92 | 20.89 | 21.40 | 18.69 |
| S3PO-GS | 17.98 | 19.54 | 20.27 | 21.24 | 21.20 | 20.62 |
| OnTheFly-NVS | 14.88 | 16.33 | 14.66 | 16.37 | 15.47 | 14.47 |
| LongSplat | - | 19.52 | 19.01 | 19.71 | 18.94 | 19.17 |
| Ours | **23.28** | **24.74** | **26.34** | **23.07** | **21.80** | **25.37** |

Table 15: SSIM on the ScanNet Dataset

| Method | scene0000_00 | scene0059_00 | scene0106_00 | scene0169_00 | scene0181_00 | scene0207_00 |
|---|---|---|---|---|---|---|
| MonoGS | - | 0.733 | 0.774 | 0.792 | 0.823 | 0.776 |
| SEGS-SLAM | - | - | 0.804 | 0.850 | **0.902** | 0.799 |
| S3PO-GS | 0.738 | 0.769 | 0.816 | 0.815 | 0.845 | 0.797 |
| OnTheFly-NVS | 0.742 | 0.747 | 0.667 | 0.721 | 0.667 | 0.705 |
| LongSplat | - | 0.752 | 0.764 | 0.750 | 0.782 | 0.721 |
| Ours | **0.824** | **0.863** | **0.905** | **0.859** | 0.889 | **0.848** |

Table 16: LPIPS on the ScanNet Dataset

| Method | scene0000_00 | scene0059_00 | scene0106_00 | scene0169_00 | scene0181_00 | scene0207_00 |
|---|---|---|---|---|---|---|
| MonoGS | - | 0.713 | 0.597 | 0.612 | 0.578 | 0.644 |
| SEGS-SLAM | - | - | 0.405 | 0.352 | **0.269** | 0.432 |
| S3PO-GS | 0.700 | 0.518 | 0.484 | 0.550 | 0.513 | 0.583 |
| OnTheFly-NVS | 0.464 | 0.477 | 0.554 | 0.504 | 0.491 | 0.474 |
| LongSplat | - | 0.402 | 0.384 | 0.384 | 0.423 | 0.427 |
| Ours | **0.254** | **0.279** | **0.237** | **0.278** | 0.288 | **0.290** |

Table 17: PSNR on the Waymo Dataset

| Method | 100613 | 106762 | 132384 | 13476 | 152706 | 153495 | 158686 | 163453 | 405841 |
|---|---|---|---|---|---|---|---|---|---|
| MonoGS | 20.05 | 20.91 | 22.71 | 19.51 | 21.23 | 14.15 | 20.29 | 19.01 | 16.19 |
| SEGS-SLAM | 20.14 | 23.60 | 22.52 | - | 24.11 | 21.16 | 21.22 | - | 19.01 |
| S3PO-GS | 25.36 | 28.23 | 27.01 | 24.85 | 28.53 | 26.54 | 26.03 | 23.65 | 27.28 |
| OnTheFly-NVS | 26.95 | 27.21 | 25.31 | 24.34 | 25.45 | 26.41 | 26.30 | 23.97 | 23.79 |
| LongSplat | 24.25 | 23.70 | 24.11 | 24.42 | 25.66 | 23.74 | 24.84 | 22.69 | 25.61 |
| Ours | **27.98** | **30.84** | **30.32** | **28.03** | **29.80** | **27.60** | **26.83** | **26.91** | **30.48** |

Table 18: SSIM on the Waymo Dataset

| Method | 100613 | 106762 | 132384 | 13476 | 152706 | 153495 | 158686 | 163453 | 405841 |
|---|---|---|---|---|---|---|---|---|---|
| MonoGS | 0.758 | 0.816 | 0.855 | 0.713 | 0.792 | 0.672 | 0.723 | 0.745 | 0.695 |
| SEGS-SLAM | 0.730 | 0.805 | 0.824 | - | 0.784 | 0.734 | 0.677 | - | 0.698 |
| S3PO-GS | 0.828 | 0.878 | 0.883 | 0.778 | 0.856 | 0.846 | 0.819 | 0.797 | 0.865 |
| OnTheFly-NVS | 0.850 | 0.854 | 0.864 | 0.764 | 0.795 | 0.847 | 0.837 | 0.785 | 0.788 |
| LongSplat | 0.773 | 0.766 | 0.822 | 0.716 | 0.778 | 0.755 | 0.777 | 0.732 | 0.795 |
| Ours | **0.865** | **0.906** | **0.919** | **0.856** | **0.882** | **0.871** | **0.847** | **0.866** | **0.907** |

Table 19: LPIPS on the Waymo Dataset

| Method | 100613 | 106762 | 132384 | 13476 | 152706 | 153495 | 158686 | 163453 | 405841 |
|---|---|---|---|---|---|---|---|---|---|
| MonoGS | 0.610 | 0.534 | 0.451 | 0.745 | 0.666 | 0.710 | 0.627 | 0.664 | 0.633 |
| SEGS-SLAM | 0.484 | 0.399 | 0.384 | - | 0.450 | 0.472 | 0.495 | - | 0.502 |
| S3PO-GS | 0.329 | 0.276 | 0.275 | 0.471 | 0.427 | 0.379 | 0.373 | 0.411 | 0.352 |
| OnTheFly-NVS | 0.328 | 0.337 | 0.361 | 0.378 | 0.401 | 0.348 | 0.307 | 0.376 | 0.400 |
| LongSplat | 0.356 | 0.328 | 0.355 | 0.354 | 0.397 | 0.430 | 0.321 | 0.371 | 0.326 |
| Ours | **0.308** | **0.237** | **0.265** | **0.267** | **0.304** | **0.313** | **0.283** | **0.289** | **0.216** |

Table 20: PSNR on the VR-NeRF Dataset

| Method | appartment262 | kitchen261 | kitchen262 | kitchen263 | table61 | workspace61 | workspace62 | workspace64 |
|---|---|---|---|---|---|---|---|---|
| MonoGS | 18.43 | 16.91 | 11.66 | 14.47 | 15.50 | 14.80 | 15.12 | 14.80 |
| SEGS-SLAM | 26.14 | **31.81** | - | **32.65** | **36.55** | **30.95** | - | - |
| S3PO-GS | 28.45 | 27.98 | 25.16 | 18.56 | 19.36 | 22.16 | 23.49 | 22.17 |
| LongSplat | 31.22 | 27.10 | 27.76 | 24.04 | 24.90 | - | 22.32 | 22.84 |
| OnTheFly-NVS | 30.52 | 30.24 | 27.51 | 25.16 | 27.77 | 22.08 | **26.04** | **29.05** |
| Ours | **32.98** | 30.90 | **30.23** | 29.04 | 29.05 | 24.68 | 24.63 | 27.13 |

Table 21: SSIM on the VR-NeRF Dataset

| Method | appartment262 | kitchen261 | kitchen262 | kitchen263 | table61 | workspace61 | workspace62 | workspace64 |
|---|---|---|---|---|---|---|---|---|
| MonoGS | 0.646 | 0.627 | 0.506 | 0.599 | 0.595 | 0.526 | 0.581 | 0.522 |
| SEGS-SLAM | 0.831 | 0.910 | - | 0.883 | **0.949** | **0.905** | - | - |
| S3PO-GS | 0.875 | 0.880 | 0.856 | 0.719 | 0.737 | 0.762 | 0.791 | 0.758 |
| LongSplat | 0.905 | 0.861 | 0.888 | 0.803 | 0.823 | - | 0.757 | 0.787 |
| OnTheFly-NVS | 0.912 | 0.903 | 0.900 | 0.847 | 0.882 | 0.775 | **0.855** | **0.898** |
| Ours | **0.937** | **0.913** | **0.939** | **0.913** | 0.900 | 0.842 | 0.833 | 0.883 |

Table 22: LPIPS on the VR-NeRF Dataset

| Method | appartment262 | kitchen261 | kitchen262 | kitchen263 | table61 | workspace61 | workspace62 | workspace64 |
|---|---|---|---|---|---|---|---|---|
| MonoGS | 0.631 | 0.688 | 0.676 | 0.662 | 0.582 | 0.689 | 0.721 | 0.685 |
| SEGS-SLAM | 0.375 | **0.218** | - | **0.192** | **0.174** | **0.203** | - | - |
| S3PO-GS | 0.321 | 0.302 | 0.269 | 0.597 | 0.474 | 0.385 | 0.350 | 0.386 |
| LongSplat | 0.292 | 0.308 | 0.265 | 0.319 | 0.353 | - | 0.359 | 0.352 |
| OnTheFly-NVS | 0.302 | 0.277 | 0.311 | 0.311 | 0.322 | 0.385 | **0.307** | **0.261** |
| Ours | **0.201** | 0.224 | **0.185** | 0.198 | 0.256 | 0.286 | 0.314 | 0.274 |

Table 23: PSNR on the KITTI Dataset

| Method | 00 | 02 | 03 | 05 | 06 | 07 | 08 | 10 |
|---|---|---|---|---|---|---|---|---|
| MonoGS | 16.01 | 15.08 | 16.90 | 15.66 | 16.38 | 11.38 | 13.21 | 11.82 |
| SEGS-SLAM | 12.69 | 14.88 | 16.71 | 14.88 | 14.65 | 10.45 | 14.50 | 13.51 |
| S3PO-GS | _20.77_ | _19.30_ | _20.51_ | _20.73_ | _20.42_ | _20.38_ | _20.27_ | _17.41_ |
| OnTheFly-NVS | 15.97 | 17.00 | 17.34 | 18.08 | 17.29 | 18.21 | 16.50 | 14.73 |
| LongSplat | 17.84 | 14.62 | 17.97 | 18.08 | 18.68 | 16.14 | 16.58 | 14.93 |
| Ours | **23.76** | **22.53** | **24.54** | **23.80** | **23.59** | **23.92** | **22.86** | **20.38** |

Table 24: SSIM on the KITTI Dataset

| Method | 00 | 02 | 03 | 05 | 06 | 07 | 08 | 10 |
|---|---|---|---|---|---|---|---|---|
| MonoGS | 0.568 | 0.476 | 0.487 | 0.491 | 0.560 | 0.439 | 0.480 | 0.420 |
| SEGS-SLAM | 0.454 | 0.458 | 0.448 | 0.470 | 0.510 | 0.405 | 0.493 | 0.462 |
| S3PO-GS | _0.732_ | _0.587_ | _0.581_ | _0.659_ | _0.652_ | _0.715_ | _0.684_ | _0.545_ |
| OnTheFly-NVS | 0.594 | 0.538 | 0.529 | 0.606 | 0.583 | 0.673 | 0.622 | 0.483 |
| LongSplat | 0.616 | 0.444 | 0.482 | 0.548 | 0.583 | 0.557 | 0.543 | 0.484 |
| Ours | **0.829** | **0.707** | **0.745** | **0.781** | **0.779** | **0.827** | **0.786** | **0.663** |

Table 25: LPIPS on the KITTI Dataset

| Method | 00 | 02 | 03 | 05 | 06 | 07 | 08 | 10 |
|---|---|---|---|---|---|---|---|---|
| MonoGS | 0.687 | 0.761 | 0.735 | 0.753 | 0.720 | 0.826 | 0.830 | 0.820 |
| SEGS-SLAM | 0.492 | 0.491 | 0.465 | 0.453 | 0.464 | 0.552 | 0.450 | 0.533 |
| S3PO-GS | _0.264_ | _0.459_ | 0.498 | _0.389_ | 0.401 | _0.364_ | _0.345_ | 0.563 |
| OnTheFly-NVS | 0.461 | 0.501 | 0.501 | 0.455 | 0.500 | 0.388 | 0.422 | 0.540 |
| LongSplat | 0.375 | 0.531 | _0.438_ | 0.428 | _0.394_ | 0.454 | 0.440 | _0.517_ |
| Ours | **0.234** | **0.375** | **0.311** | **0.281** | **0.288** | **0.242** | **0.266** | **0.394** |

Table 26: PSNR on the ScanNet++ Dataset

| Method | 00777c41d4 | 02f25e5fee | 0b031f3119 | 126d03d821 | 1cbb105c6a | 2284bf5c9d | 2d2e873aa0 |
|---|---|---|---|---|---|---|---|
| MonoGS | 14.621 | 20.915 | 9.488 | 21.353 | 23.716 | 13.049 | 16.861 |
| SEGS-SLAM | - | _28.177_ | - | - | - | - | - |
| S3PO-GS | _21.316_ | 25.158 | 21.019 | 23.758 | 26.614 | _23.446_ | _23.554_ |
| OnTheFly-NVS | 16.270 | 19.134 | 21.701 | 17.572 | 17.478 | 18.200 | 14.173 |
| LongSplat | 18.465 | 26.747 | _22.484_ | _27.251_ | _28.621_ | 23.348 | - |
| Ours | **26.543** | **30.671** | **27.722** | **32.796** | **32.406** | **30.277** | **28.137** |

| Method | 303745abc7 | 41eb967018 | 46001f434d | 4808c4a397 | 546292a9db | 712dc47104 | 7543973e1a |
|---|---|---|---|---|---|---|---|
| MonoGS | 14.899 | 18.011 | 18.515 | 21.570 | 9.464 | 8.970 | 22.547 |
| SEGS-SLAM | - | - | - | - | - | - | 27.169 |
| S3PO-GS | _25.840_ | _21.074_ | 20.288 | _24.263_ | 20.612 | _17.843_ | 26.367 |
| OnTheFly-NVS | 15.884 | 16.247 | 16.426 | 18.722 | _24.791_ | 14.056 | 21.540 |
| LongSplat | - | - | **24.437** | - | 24.587 | - | _28.539_ |
| Ours | **33.056** | **28.608** | _21.716_ | **30.101** | **25.868** | **27.506** | **32.308** |

Table 27: SSIM on the ScaNnet++ Dataset

| Method | 00777c41d4 | 02f25e5fee | 0b031f3119 | 126d03d821 | 1cbb105c6a | 2284bf5c9d | 2d2e873aa0 |
|---|---|---|---|---|---|---|---|
| MonoGS | 0.559 | 0.784 | 0.485 | 0.796 | 0.833 | 0.622 | 0.641 |
| SEGS-SLAM | - | 0.911 | - | - | - | - | - |
| S3PO-GS | 0.695 | 0.853 | 0.809 | 0.828 | 0.876 | 0.833 | 0.830 |
| OnTheFly-NVS | 0.583 | 0.779 | 0.823 | 0.753 | 0.749 | 0.751 | 0.705 |
| LongSplat | 0.604 | 0.868 | 0.816 | 0.854 | 0.897 | 0.819 | - |
| Ours | **0.855** | **0.941** | **0.903** | **0.942** | **0.956** | **0.937** | **0.918** |

| Method | 303745abc7 | 41eb967018 | 46001f434d | 4808c4a397 | 546292a9db | 712dc47104 | 7543973e1a |
|---|---|---|---|---|---|---|---|
| MonoGS | 0.723 | 0.737 | 0.844 | 0.848 | 0.244 | 0.590 | 0.836 |
| SEGS-SLAM | - | - | - | - | - | - | 0.891 |
| S3PO-GS | 0.878 | 0.788 | 0.861 | 0.873 | 0.688 | 0.784 | 0.884 |
| OnTheFly-NVS | 0.780 | 0.705 | 0.817 | 0.826 | 0.825 | 0.732 | 0.831 |
| LongSplat | - | - | **0.900** | - | 0.793 | - | 0.894 |
| Ours | **0.959** | **0.905** | 0.891 | **0.938** | **0.850** | **0.915** | **0.950** |

Table 28: LPIPS on the ScaNet++ Dataset

| Method | 00777c41d4 | 02f25e5fee | 0b031f3119 | 126d03d821 | 1cbb105c6a | 2284bf5c9d | 2d2e873aa0 |
|---|---|---|---|---|---|---|---|
| MonoGS | 0.822 | 0.459 | 0.724 | 0.450 | 0.370 | 0.710 | 0.769 |
| SEGS-SLAM | - | 0.148 | - | - | - | - | - |
| S3PO-GS | 0.451 | 0.243 | 0.350 | 0.346 | 0.264 | 0.289 | 0.316 |
| OnTheFly-NVS | 0.524 | 0.352 | 0.310 | 0.404 | 0.409 | 0.386 | 0.499 |
| LongSplat | 0.436 | 0.179 | 0.307 | 0.242 | 0.179 | 0.245 | - |
| Ours | **0.209** | **0.122** | **0.196** | **0.143** | **0.123** | **0.134** | **0.179** |

| Method | 303745abc7 | 41eb967018 | 46001f434d | 4808c4a397 | 546292a9db | 712dc47104 | 7543973e1a |
|---|---|---|---|---|---|---|---|
| MonoGS | 0.707 | 0.619 | 0.485 | 0.408 | 0.734 | 0.707 | 0.441 |
| SEGS-SLAM | - | - | - | - | - | - | 0.216 |
| S3PO-GS | 0.259 | 0.454 | 0.419 | 0.300 | 0.520 | 0.449 | 0.303 |
| OnTheFly-NVS | 0.382 | 0.434 | 0.362 | 0.331 | 0.256 | 0.450 | 0.305 |
| LongSplat | - | - | **0.265** | - | 0.286 | - | 0.200 |
| Ours | **0.115** | **0.165** | 0.279 | **0.156** | **0.205** | **0.176** | **0.140** |

Table 29: Ablations for Localization and Reconstruction

| Method | 0077 | 02f2 | 0b03 | 126d | 1cbb | 2284 | 2d2e | 3037 | 41eb | 4600 | 4808 | 5462 | 712d | 7543 | Avg. |
|---|---|---|---|---|---|---|---|---|---|---|---|---|---|---|---|
| w/ $\pi^3$ | 0.305 | 0.192 | 0.576 | 0.310 | 0.153 | 0.320 | - | 0.582 | 0.582 | - | - | 0.140 | 0.557 | 0.393 | 0.374 |
| w/ loop (vggt) | 0.010 | **0.011** | 0.014 | 1.176 | 0.014 | 0.022 | **0.010** | 0.007 | 0.030 | 0.075 | 0.022 | 0.032 | **0.013** | 0.011 | 0.096 |
| w/o loop | 0.019 | 0.017 | 0.015 | **0.016** | 0.011 | 0.012 | **0.010** | 0.290 | 0.032 | **0.052** | 0.066 | 0.230 | **0.013** | 0.011 | 0.057 |
| Track w/ MF&KF | 0.011 | 0.027 | **0.012** | 1.050 | **0.010** | **0.008** | 0.014 | **0.005** | 0.022 | 0.076 | 0.040 | **0.024** | **0.013** | **0.010** | 0.094 |
| Ours full model | **0.009** | **0.011** | 0.015 | 0.019 | 0.015 | 0.012 | **0.010** | **0.005** | **0.017** | 0.060 | **0.021** | 0.030 | 0.014 | 0.011 | **0.018** |

Table 30: Tracking Results on the ScanNet++ Dataset

| Method | 0077 | 02f2 | 0b03 | 126d | 1cbb | 2284 | 2d2e | 3037 | 41eb | 4600 | 4808 | 5462 | 712d | 7543 |
|---|---|---|---|---|---|---|---|---|---|---|---|---|---|---|
| OnTheFly | 0.804 | 0.157 | 0.114 | 0.227 | 0.227 | 0.913 | 1.438 | 1.186 | 1.605 | 3.932 | 0.486 | **0.018** | 0.458 | 0.907 |
| LongSplat | 1.307 | 0.041 | 0.378 | 0.550 | 0.023 | 0.713 | - | - | - | 1.943 | - | 0.408 | - | 0.058 |
| MonoGS | 1.364 | 0.626 | - | - | 0.294 | 0.939 | 2.019 | 0.922 | 1.760 | 3.540 | 0.432 | - | 0.853 | 0.642 |
| S3PO-GS | 0.650 | 0.016 | 0.093 | 0.418 | 0.135 | 0.441 | 0.355 | 0.858 | 1.674 | 1.318 | 0.614 | 0.909 | 1.092 | 0.270 |
| SEGS-SLAM | - | **0.005** | - | - | - | - | - | - | - | - | - | - | - | 0.485 |
| MASt3R-SLAM | 0.121 | 0.017 | 0.059 | 0.122 | 0.021 | 0.027 | 0.835 | 0.176 | 0.020 | 2.379 | 0.065 | 0.042 | 0.021 | 0.025 |
| loop with vggt | 0.010 | 0.011 | **0.014** | 1.176 | **0.014** | 0.022 | 0.010 | 0.007 | 0.030 | 0.075 | 0.022 | 0.032 | **0.013** | **0.011** |
| Ours | **0.009** | 0.011 | 0.015 | 0.019 | 0.015 | 0.012 | **0.010** | **0.005** | **0.017** | 0.060 | **0.021** | 0.030 | 0.014 | **0.011** |

Table 31: Tracking Results on the TUM Dataset

| Method | f1_360 | f1_desk | f1_desk2 | f1_floor | f1_plant | f1_room | f1_rpy | f1_teddy | f1_xyz | f2_xyz | f3_office |
|---|---|---|---|---|---|---|---|---|---|---|---|
| OnTheFly | 0.187 | - | - | - | - | 0.874 | 4.136 | - | 0.278 | 0.073 | 1.489 |
| LongSplat | 0.133 | - | - | - | - | 0.712 | - | - | - | 0.103 | - |
| MonoGS | 0.160 | 0.034 | 0.596 | 0.531 | 0.077 | 0.649 | 0.032 | 0.512 | 0.017 | 0.047 | 0.033 |
| S3PO-GS | 0.093 | 0.041 | 0.155 | 0.228 | 0.040 | 0.552 | 0.053 | 0.051 | 0.010 | 0.016 | 0.045 |
| SEGS-SLAM | 0.154 | **0.016** | **0.013** | - | - | - | - | 0.293 | 0.009 | 0.006 | 0.026 |
| MASt3R-SLAM | 0.049 | **0.016** | 0.024 | **0.025** | 0.020 | 0.061 | 0.027 | **0.041** | 0.009 | 0.005 | 0.031 |
| loop with vggt | **0.040** | **0.016** | 0.026 | 0.026 | 0.017 | **0.056** | 0.023 | 0.047 | **0.007** | **0.005** | 0.024 |
| Ours | **0.040** | **0.016** | 0.025 | **0.025** | 0.016 | 0.060 | **0.022** | 0.045 | **0.007** | **0.005** | **0.019** |

Table 32: Tracking Results on the KITTI Dataset

| Method | 00 | 02 | 03 | 05 | 06 | 07 | 08 | 10 | Avg. |
|---|---|---|---|---|---|---|---|---|---|
| OnTheFly | 18.297 | 28.230 | 14.356 | 5.027 | 9.608 | 1.550 | 9.836 | 9.502 | 12.051 |
| LongSplat | 3.047 | 10.796 | 9.344 | 9.482 | 6.864 | 2.866 | 4.480 | 6.576 | 6.682 |
| MonoGS | 11.868 | 11.817 | 11.195 | 4.623 | 7.638 | 4.128 | 5.864 | 5.109 | 7.780 |
| S3PO-GS | 1.196 | 2.961 | 5.522 | 1.459 | **0.721** | 1.009 | 2.655 | 1.927 | 2.181 |
| SEGS-SLAM | **0.561** | **0.897** | **0.189** | 0.967 | 4.122 | **0.851** | **0.477** | **0.353** | **1.052** |
| MASt3R-SLAM | - | 1.756 | 0.397 | **0.761** | 2.279 | 1.361 | - | - | – |
| loop with vggt | 1.304 | 2.691 | 0.442 | 1.103 | 0.958 | 1.257 | 1.160 | 1.893 | 1.351 |
| Ours | 1.304 | 2.691 | 0.442 | 1.103 | 0.958 | 1.321 | 1.167 | 1.893 | 1.360 |

Table 33: Tracking Results on the Waymo Dataset

| Method | 13476 | 100613 | 106762 | 132384 | 152706 | 153495 | 158686 | 163453 | 405841 | Avg. |
|---|---|---|---|---|---|---|---|---|---|---|
| OnTheFly | 1.687 | 1.051 | 2.022 | 14.395 | 1.193 | 2.110 | 2.206 | 1.872 | 1.523 | 3.118 |
| LongSplat | 7.024 | 10.242 | 7.506 | 2.616 | 4.900 | 2.625 | 3.921 | 3.939 | 2.729 | 4.956 |
| MonoGS | 3.262 | 6.706 | 15.638 | 11.093 | 8.286 | **0.585** | 6.881 | 10.669 | 3.218 | 7.370 |
| S3PO-GS | **1.165** | 2.471 | **0.214** | **1.551** | 1.144 | 1.972 | 0.963 | 1.085 | **0.560** | 1.236 |
| SEGS-SLAM | - | 0.860 | 0.696 | 2.755 | 1.867 | - | - | - | - | - |
| MASt3R-SLAM | - | - | 2.958 | - | - | 1.569 | 1.098 | 2.334 | 0.873 | - |
| loop with vggt | 1.230 | **0.315** | 2.762 | 1.753 | **0.931** | 1.578 | **0.460** | 1.071 | 0.816 | **1.213** |
| Ours | 1.229 | **0.315** | 2.762 | 1.753 | **0.931** | 1.578 | **0.460** | 1.071 | 0.816 | **1.213** |

Table 34: Tracking Results on the TUM Dataset Compared with Non 3DGS SLAM Systems

| Method | 360 | desk | desk2 | floor | plant | room | rpy | teddy | xyz | Avg. |
|---|---|---|---|---|---|---|---|---|---|---|
| ORB-SLAM3 | - | 0.017 | 0.210 | - | 0.034 | - | - | - | 0.009 | - |
| DPV-SLAM++ | 0.132 | 0.018 | 0.029 | 0.050 | 0.022 | 0.096 | 0.032 | 0.098 | 0.010 | 0.054 |
| DROID-SLAM | 0.111 | 0.018 | 0.042 | **0.021** | **0.016** | **0.049** | 0.026 | 0.048 | 0.012 | 0.038 |
| Go-SLAM | 0.089 | **0.016** | 0.028 | 0.025 | 0.026 | 0.052 | **0.019** | 0.048 | 0.010 | 0.035 |
| MASt3R-SLAM | 0.049 | **0.016** | **0.024** | 0.025 | 0.020 | 0.061 | 0.027 | **0.041** | 0.009 | 0.030 |
| Ours | **0.040** | **0.016** | 0.025 | 0.025 | **0.016** | 0.060 | 0.022 | 0.045 | **0.007** | **0.028** |

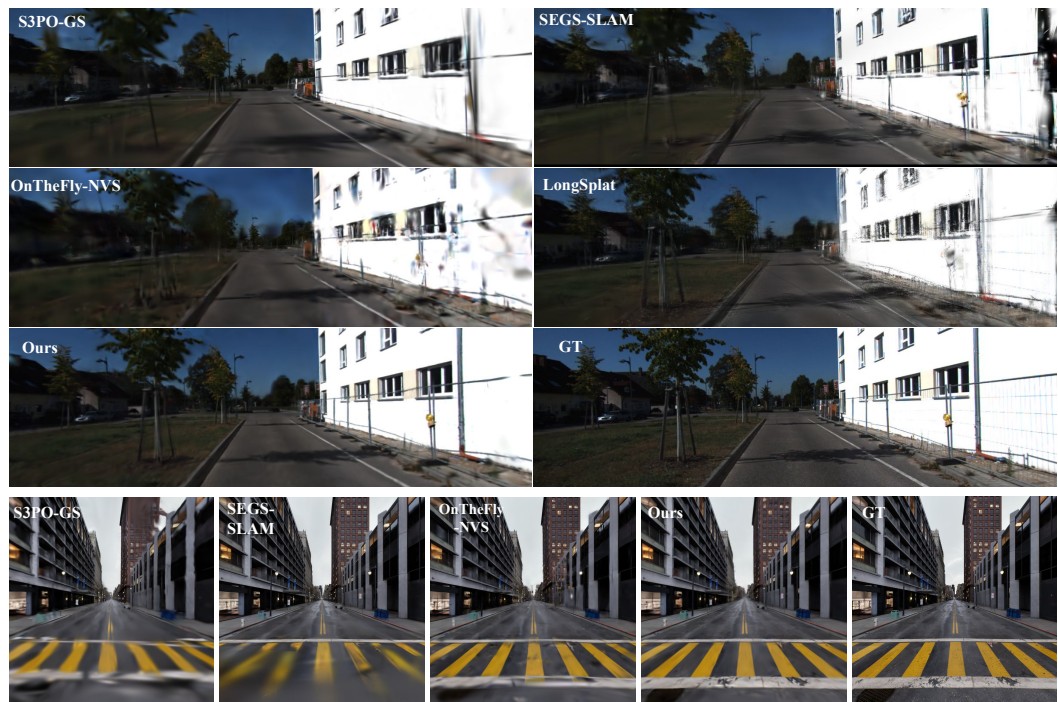

Figure 5: More Qualitative Reconstruction Results.

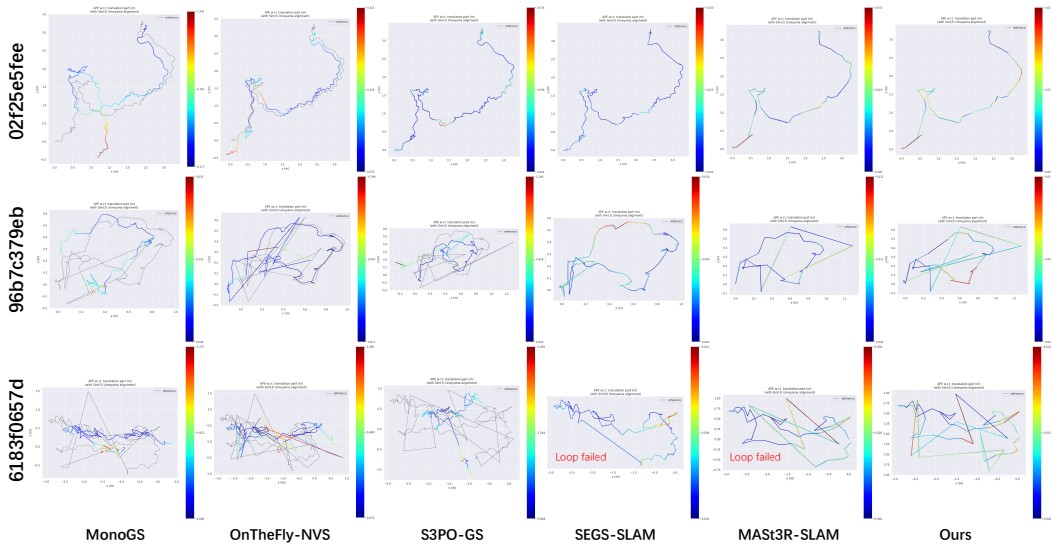

Figure 6: Qualitative Comparison of Trajectories across Different Methods on the ScanNet++ Dataset.

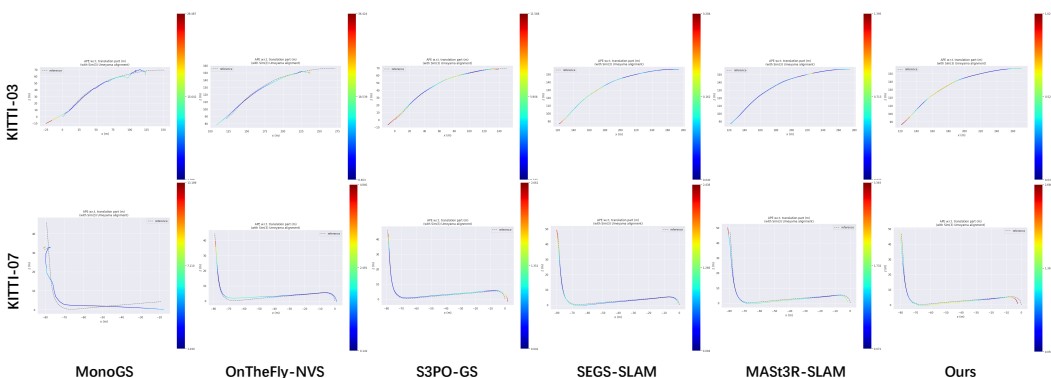

Figure 7: Qualitative Comparison of Trajectories across Different Methods on the KITTI Dataset.

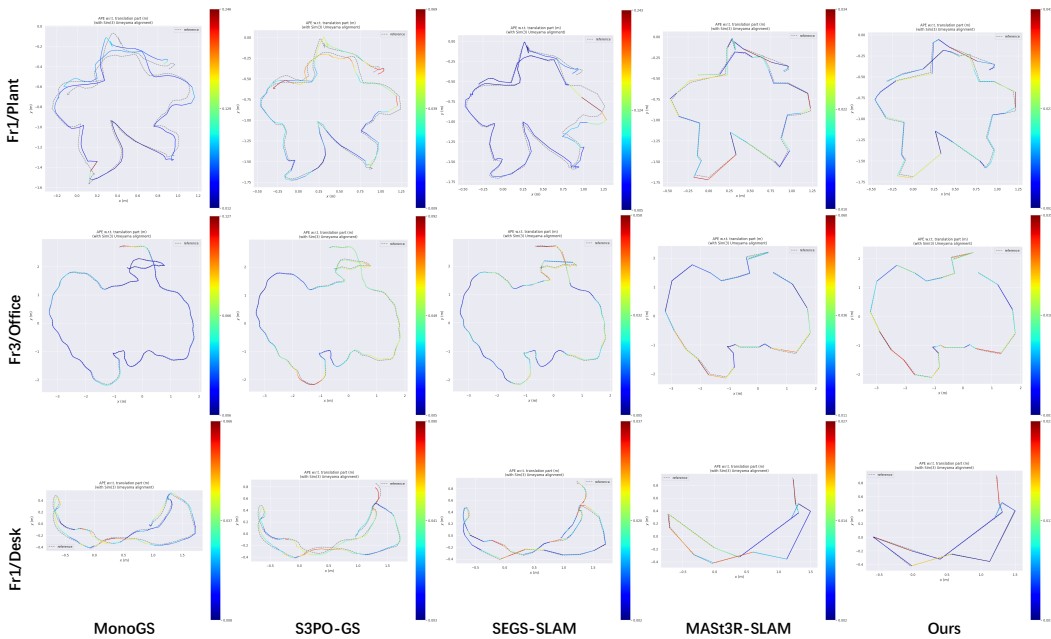

Figure 8: Qualitative Comparison of Trajectories across Different Methods on the TUM Dataset.

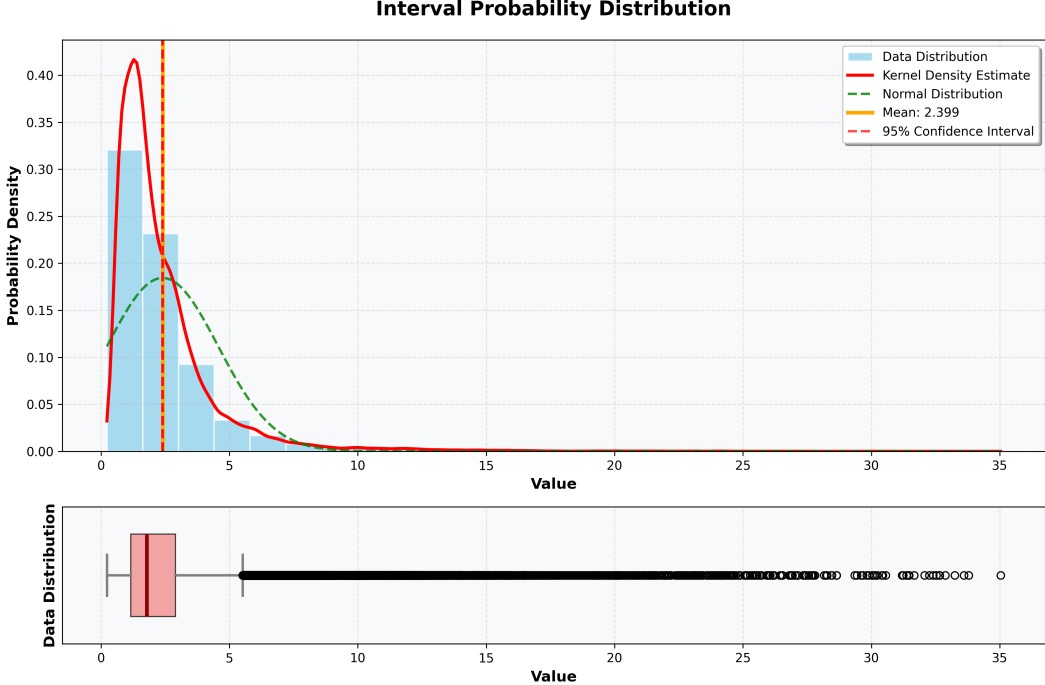

Figure 9: Visualization of the distribution of the $d_{max}$ attribute on ScanNet++ Yeshwanth et al. (2023) datasets.

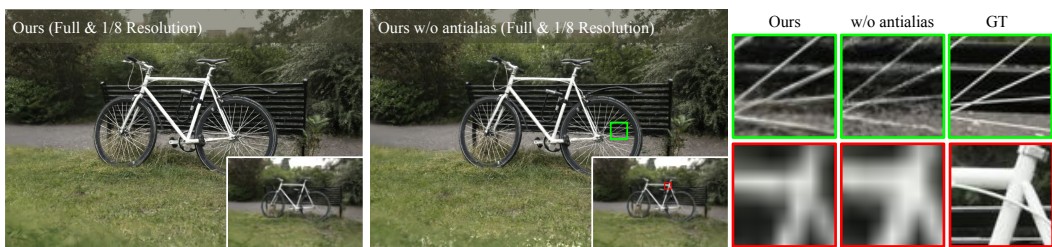

Figure 10: Qualitative comparison of full-resolution and low-resolution (1/8 of full-resolution) on multi-resolution Mip-NeRF360 Barron et al. (2022) datasets.

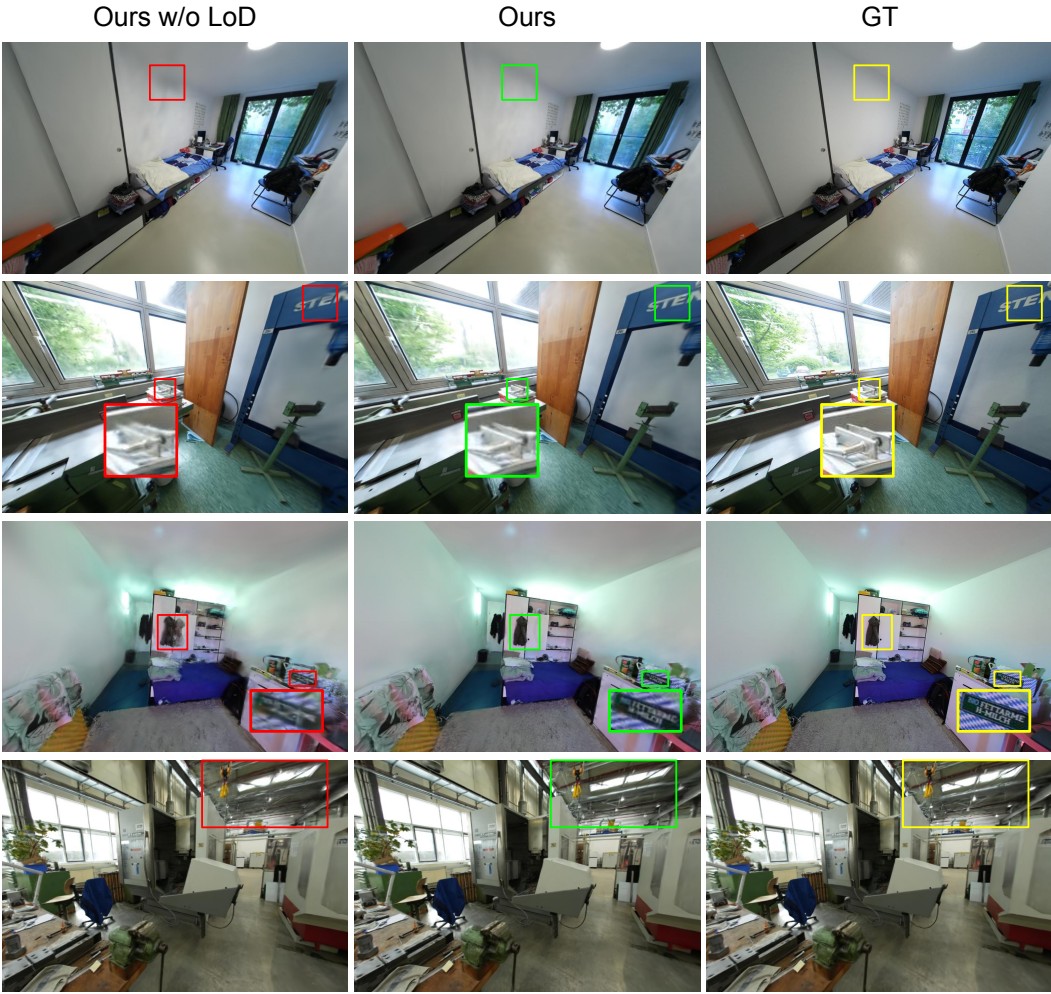

Figure 11: Qualitative Comparison of the ablation study on level-of-detail on ScanNet++ Yeshwanth et al. (2023) datasets.

