# OpenReview forum: "ARTDECO: Toward High-Fidelity On-the-Fly Reconstruction with Hierarchical Gaussian Structure and Feed-Forward Guidance"
_ICLR.cc/2026/Conference — ICLR 2026 Poster_

### Official Review · Reviewer_Wwfj · 2025-10-27

**Soundness:** 3
**Presentation:** 3
**Contribution:** 3
**Rating:** 6
**Confidence:** 5

**Summary:**

The paper introduces a unified framework for on-the-fly, high-fidelity 3D reconstruction from monocular image sequences. It addresses the longstanding tradeoff between efficiency, accuracy, and robustness in real-time 3D scene capture by combining the strengths of feed-forward 3D foundation models and SLAM-style geometric optimization, using 3D Gaussian Splatting (3DGS) as the scene representation.

**Strengths:**

The paper is clearly written and well-structured. The problem statement (monocular on-the-fly reconstruction) and core tradeoff (efficiency vs. accuracy vs. robustness) are crisply articulated in the introduction. Figures 2 and 3 provide intuitive visual summaries of the pipeline and LoD mechanism, respectively.

**Weaknesses:**

1. The paper describes three modules (frontend, backend, mapping) but lacks clarity on how tightly coupled they are and whether components can be independently replaced or ablated. For example:
  - The frontend uses MASt3R for pose estimation, but the backend uses π3 for loop closure. Why not use $\pi_3$ in both? The ablation (Table 3) shows $\pi_3$ underperforms in pose estimation, but the paper doesn’t explain why this is the case (e.g., is it due to training data, architecture, or metric scale ambiguity?).
  - The mapping module relies on pointmaps from the backend, yet it’s unclear how errors in pose or loop closure propagate into Gaussian initialization.

2. The paper compares against LongSplat and OnTheFly-NVS, but omits very recent pose-free methods like AnySplat or No Pose, No Problem in key metrics. More critically, the evaluation protocol (every 8th frame held out) is applied retroactively to baselines, but it’s unclear if those methods were designed for such a protocol. For example:
 - Feed-forward methods often assume all input frames are available at once; streaming evaluation may disadvantage them unfairly.
 - Conversely, SLAM methods like MASt3R-SLAM are inherently streaming—so the comparison may favor ARTDECO by design.

3. The core claim is that ARTDECO “combines the efficiency of feed-forward models with the reliability of SLAM.” However, the paper does not quantify how much of the performance gain comes from:
 - The foundation models (MASt3R/$\pi_3$),
 - The LoD Gaussian representation,
 - The SLAM-style optimization (BA, loop closure).

The ablation in Table 3 is helpful but incomplete. For instance:
 - There is no ablation removing both loop closure and mapper frames.
 - The “w/o level-of-detail” ablation shows a PSNR drop (~1 dB), but it’s unclear whether this is due to rendering artifacts, geometry errors, or memory inefficiency.

**Questions:**

See the weaknesses.

---

> ### Author Response · Authors · 2025-11-24
> **Point-to-Point Response to Reviewer Wwfj (1/2)**
>
> We sincerely thank Reviewer Wwfj for the valuable comments on the paper's logic, experimental setup, and ablation studies. We have conducted experiments following the relevant suggestions, which have significantly improved the paper's logic and solidity. Below are our point-by-point responses:
>
> **W1:**
>
> Thank you for your suggestion. We indeed lack relevant analysis. Here are the explainations:
>
> (1) In the table below, we report the results of our full system using $\pi^{3}$ as the main visual backbone, and its performance is lower than MAST3R. The reasons are as follows:
>
> - $\pi^{3}$ supports multi-frame inference, multi-frame models tend to confuse relative object proportions when views differ. As a result, even after scale alignment, noticeable layering artifacts appear in the fused point cloud. MAST3R, trained with metric-scale supervision and using two-frame inference, also exhibits this issue slightly. However, in long sequences, repeated two-frame inference combined with a filtering-like aggregation (see MAST3R-SLAM Sec. 3.3, Eq. 8 and associated analysis) can mitigate this problem. One might ask whether running $\pi^{3}$ in a single two-frame inference pass would match MAST3R. We performed this experiment and found that $\pi^{3}$ performs poorly when only a few frames are available, likely because it was primarily trained for multi-frame scenarios.
>
> - Compared to MAST3R, $\pi^{3}$ lacks an inter-image pixel correspondence head, which is crucial for subsequent pose refinement. The ablation study in MAST3R-SLAM (Table 4) confirms the importance of this component.
>
> Comparision Between MASt3R and $\pi^{3}$
>
> | Methods      | Tracking | PSNR  | SSIM  | LPIPS |
> | ------------ | -------- | ----- | ----- | ----- |
> | $\pi^{3}$+BA | 0.374    | 15.46 | 0.713 | 0.438 |
> | Ours         | 0.018    | 29.12 | 0.918 | 0.167 |
>
>
> Ablation From MASt3R-SLAM (Table 4)
>
> | Method                            | ATE RMSE (with calib) |
> | --------------------------------- | --------------------- |
> | Point Matching                    | 0.062                 |
> | Point Matching + Feature Matching | 0.039                 |
>
>
> For the reasons above, we ultimately chose MASt3R as the main model. However, we observed that $\pi^{3}$ can produce a highly reasonable scene in a single inference pass: even if some object proportions are incorrect, the results still maintain consistency in intrinsics, extrinsics, and point clouds. Based on this observation, we leverage $\pi^{3}$’s single-pass inference capability for loop-closure detection. $\pi^{3}$-based loop closure infers the corresponding point clouds of all relevant pixels in a unified coordinate frame, allowing frame-to-frame overlap to be naturally determined through **explicit 3D point-cloud overlap**. Unlike 2D feature or bag-of-words approaches, this direct 3D overlap estimation is more accurate and provides a clear and intuitive explanation as well as insightful understanding of scene correspondence.
>
> Overall, MASt3R provides **stable, metric, two-frame geometry** for streaming tracking, while $\pi^{3}$ provides **global, single-pass 3D consistency** ideal for loop closure. The two modules are complementary rather than equally interchangeable.
>
> (2) Before initializing Gaussians using the point map from the backend, we first filter the point map based on reprojection residuals. Specifically, points are projected onto neighboring keyframes, and those with high reprojection errors are excluded from Gaussian initialization. This prevents errors from BA or loop closure from propagating into the 3DGS map.
>
> [1] Murai, R., Dexheimer, E., & Davison, A. J. (2025). MASt3R-SLAM: Real-time dense SLAM with 3D reconstruction priors. In _Proceedings of the Computer Vision and Pattern Recognition Conference_ (pp. 16695-16705).
>
> [2]. Wang, Y., Zhou, J., Zhu, H., Chang, W., Zhou, Y., Li, Z., ... & He, T. (2025). $\pi^ 3$: Permutation-Equivariant Visual Geometry Learning. _arXiv preprint arXiv:2507.13347_.

---

> ### Author Response · Authors · 2025-11-24
> **Point-to-Point Response to Reviewer Wwfj (2/2)**
>
> **W2:**
>
> Thank you for your careful reminder. We have already modified the code for all our baselines to ensure that their evaluation metrics are computed using the same approach as our system, where every eighth frame is selected for evaluation, and the evaluated frames are not involved in the reconstruction supervision but are only used for pose optimization (Detailed in Appendix A.1).
>
> Regarding the methods such as AnySplat and NopoSplat mentioned by you, they are designed to process all input frames at once, whereas our setup involves streaming input. Therefore, we did not include comparisons with these methods in the main text. In response to your feedback, we have now supplemented comparisons with the aforementioned methods under both 32-view and 100-view settings.
>
> | Dataset            | Scannetpp  |     (in-domain)      |           |           | Fast-livo2  |      (out-domain)     |           |           |
> | ------------------ | --------------------- | --------- | --------- | --------- | ----------------------- | --------- | --------- | --------- |
> | Method             | Tracking              | PSNR      | SSIM      | LPIPS     | Tracking                | PSNR      | SSIM      | LPIPS     |
> | Anysplat (32view)  | 0.082                 | 23.41     | 0.822     | **0.176** | 0.235                   | 24.21     | 0.772     | 0.198     |
> | Ours (32view)      | **0.034**             | **26.53** | **0.856** | 0.201     | **0.065**               | **27.96** | **0.853** | **0.187** |
> | Anysplat (100view) | 0.128                 | 21.37     | 0.780     | 0.227     | 0.426                   | 21.28     | 0.672     | 0.287     |
> | Ours (100view)     | **0.076**             | **24.36** | **0.841** | **0.218** | **0.270**               | **24.56** | **0.788** | **0.255** |
>
> **W3:**
>
> Thank you for your careful suggestion. We have provided the related ablation in the tables below, which indeed makes the paper more complete/comprehensive.
>
> (1) The comparision with or without Foundation Model is listed in below table.
>
> | Method             | ATE RMSE | PSNR  | SSIM  | LPIPS |
> | ------------------ | -------- | ----- | ----- | ----- |
> | On-the-Fly BA+3dgs | 0.891    | 18.01 | 0.761 | 0.386 |
> | MASt3R+3dgs        | 0.018    | 24.88 | 0.856 | 0.254 |
>
>
> OnTheFly utilizes xFeat for feature extraction and matching, combined with triangulation and bundle adjustment (BA). This approach performs poorly in scenarios with large viewpoint changes, textureless regions, or rapid motion. In contrast, Foundation Models demonstrate certain robustness in such challenging scenarios.
>
> (2) The introduction of LOD design not only reduces overhead by constraining the number of Gaussians rendered, but also such multi-scale Gaussians are better suited for modeling low-frequency and high-frequency information in the scene.
>
> (3) We have added `without Loop Closure` and  `without BA and Loop Closure` ablation. As the procedure is that firstly Loop Closure, then BA, so the result `without BA and Loop Closure` equals to `without BA`.
>
> (4) We add a new ablation study on both loop closure and mapper frames.
>
> | Methods                              | ATE RMSE | PSNR  | SSIM  | LPIPS |
> | ------------------------------------ | -------- | ----- | ----- | ----- |
> | Ours w/o level-of-detail             | 0.018    | 28.13 | 0.912 | 0.180 |
> | Ours w/o BA                          | 0.149    | 27.03 | 0.892 | 0.191 |
> | Ours w/o Loop Closure                | 0.057    | 28.18 | 0.908 | 0.178 |
> | Ours w/o Loop Closure & mapper frame | 0.057    | 25.86 | 0.879 | 0.240 |
> | Ours                                 | 0.018    | 29.12 | 0.918 | 0.167 |
>
>
> (5) In figure 11 of the supplementary material, we select four groups of images from the Scanntpp dataset to visually demonstrate the impact of LOD design on the rendering results. It can be observed that the introduction of LOD has resulted in clearer outcomes in both textured and non-textured areas.
>
> [1]. Potje, G., Cadar, F., Araujo, A., Martins, R., & Nascimento, E. R. (2024). Xfeat: Accelerated features for lightweight image matching. In _Proceedings of the IEEE/CVF Conference on Computer Vision and Pattern Recognition_ (pp. 2682-2691).

---

> > ### Comment · Reviewer_Wwfj · 2025-11-25
> >
> > Thank you for addressing my concerns. I'm pleased to change my rating to accept.

---

> > > ### Author Response · Authors · 2025-11-25
> > >
> > > Thank you for taking the time to review our work and for your thoughtful comments. We sincerely appreciate your updated rating.

---

### Official Review · Reviewer_aQRV · 2025-10-28

**Soundness:** 2
**Presentation:** 3
**Contribution:** 2
**Rating:** 4
**Confidence:** 5

**Summary:**

The paper proposes ARTDECO, a unified online reconstruction system that integrates the feed-forward 3D model with a real-time pose estimation and loop-closure pipeline. It converts multi-view features into a hierarchical level-of-details (LoD) Gaussian scene representation and continuously refines it using all frames. This integration strikes a balance between real-time performance and reconstruction fidelity, delivering globally consistent 3D scenes with quality comparable to offline optimization.

**Strengths:**

- The paper is well structured and easy to follow. The appendix provides more detailed experimental results and implementation details.

- The work demonstrates a coherent integration of pretrained feed-forward 3D models, pose estimation, and Gaussian-based mapping into a unified online system, validated on both indoor and outdoor benchmarks.

- The hierarchical Gaussian representation and LoD-aware rendering effectively balance reconstruction fidelity and computational efficiency, achieving practical real-time performance.

**Weaknesses:**

-The engineering efforts of the proposed system is appreciated; however, the system primarily integrates existing components/concepts from previous studies, i.e., composing feed-forward 3D foundation models (MASt3R-SLAM), Gauss–Newton BA, and hierarchical Gaussian splatting, into a single pipeline. The following points offer specific comments.

- The proposed front-end tracking system is built upon MASt3R-SLAM.  It switches to a pinhole camera model + reprojection residual that is a standard bundle-adjustment design in traditional SLAM systems. This is less general than MASt3R-SLAM’s generic central camera + ray-angle residual (which is explicitly designed to be robust to depth noise and varying intrinsics). The paper does not provide analysis of failure modes under non-central cameras, zoom changes, or intrinsic effect (e.g., rolling-shutter effects), nor evidence that the chosen residual yields measurable advantages over ray-angle residuals.

- LoD introduced without proper reference and sounds incremental w.r.t. previous structured/multi-level representations.
Multiple previous works have introduced hierarchical/structured scene representations and explicit LoD mechanisms (e.g., Scaffold-GS, Hierarchical 3D Gaussians, Octree-GS, Mip-Splatting, FLoD). However, at the first occurrence of “LoD” (at Line 73), there is no clear citation for this concept in this literature. E.g., what is the difference(s) of the proposed LoD with the same concept introduced in FLoD/Octree-GS? The proposed method mainly implements a 2D patch-budget–driven coarse-to-fine scheduling (e.g., per-level pixel coverage, fixed per-level scale growth like $1.4^{2l}$, and a distance-based gating with blending). While these choices are practical, they are engineering heuristics rather than a principled spatial hierarchy. Compared to methods that anchor hierarchy in 3D space/geometry (anchors/octrees/mip anti-aliasing), the proposed 2D scheduling is arguably less principled for large-scale geometry control and view-consistent refinement/anti-aliasing. The paper does not provide a sensitivity analysis (e.g., to the 1.4 scale law, pixel-budget per level, or $d_{max}$) or an ablation against established 3D-anchored LoD schemes.

- The proposed Gaussian initialization and densification seems like reusing established solutions. The method relies on residual-driven insertion to densify Gaussians, and a decoder to map multi-scale features to structured Gaussians (including scale/rotation regularization). These mirror previous practices such as base-scale setting and anisotropic covariance regularization (e.g., Meuleman et al; Wu et al, as cited).

- Loop closure via a foundation 3D model (Pi3) is not conceptually new and appears to be a dominant factor. Using a foundation model for loop detection is not new (e.g., CLIP-based loop closure has been shown for monocular Gaussian SLAM, Tian et al). Here, Pi3 is simply plugged in as the retriever. The ablation indicates that replacing Pi3 with VGGT degrades accuracy, implying the BA stack is heavily dependent on the chosen foundation model rather than on the SLAM backend system itself. Moreover, although the paper mentions memory concerns of baselines (at Line 70), it does not report the compute/memory cost of employing Pi3 (or alternative foundation models) in the loop-closure module.

- Please check the references; there are multiple duplicates, e.g., Longsplat: Robust unposed 3d gaussian splatting for casual long videos, On-thefly reconstruction for large-scale novel view synthesis from unposed images

References:

Andreas Meuleman, Ishaan Shah, Alexandre Lanvin, Bernhard Kerbl, and George Drettakis. Onthe-fly Reconstruction for Large-Scale Novel View Synthesis from Unposed Images. ACM Transactions on Graphics, 44(4), 2025b.

Songyin Wu, Zhaoyang Lv, Yufeng Zhu, Duncan Frost, Zhengqin Li, Ling-Qi Yan, Carl Ren,
Richard Newcombe, and Zhao Dong. Monocular online reconstruction with enhanced detail
preservation

Lan, Tian, Qinwei Lin, and Haoqian Wang. "Monocular gaussian slam with language extended loop closure." arXiv preprint arXiv:2405.13748 (2024).

**Questions:**

Please refer to the questions outlined in the weakness section.

---

> ### Author Response · Authors · 2025-11-24
> **Point-to-Point Response to Reviewer aQRV (1/2)**
>
> We sincerely thank Reviewer aQRV, whose detailed comments on the paper's content, specific implementation details of the method, and typos have provided us with different perspectives and valuable suggestions for improvement. Below are our point-by-point responses:
>
> **Global Response**
>
> Thank you for your valuable comments. We would like to respectfully clarify that our key contribution lies in **system-level innovation**, where each sub-module has been explicitly redesigned to ensure compatibility, stability, and robustness under streaming input conditions. **This includes: (1) SLAM/Mapper/Common frame selection and uncertainty-aware local BA in the frontend; (2) $\pi^{3}$-based 3D-overlap loop closure with global BA and Gaussian point initialization in the backend; (3) hierarchical structured Gaussians in the mapper; (4) comprehensive evaluation across 8 datasets with open-source code. The resulting system behavior cannot be achieved by naively combining existing works.**
>
> **W1:**
>
> Our decision to use a pinhole camera model with a reprojection residual is based on the following considerations:
>
> 1. **In terms of localization accuracy, reprojection residual is better than ray-angle residual.** MASt3R-SLAM provides two optimization modes. When camera intrinsic are unknown, it uses a ray-angle residual; when intrinsic are available, it adopts a reprojection residual based on pinhole camera. The tracking results on TUM, 7-Scenes, and EuRoC of MASt3R-SLAM consistently show that reprojection residuals based on pinhole camera outperform ray-angle residuals—indicating that leveraging intrinsic parameters indeed improves accuracy.
> 2. **3DGS rendering requires intrinsic.** Most existing 3DGS pipelines adopt a pinhole-camera formulation and require intrinsic parameters, which are typically available as metadata for common streaming video data. Following these conventions and for system simplicity, we design our pipeline under the same assumption.
>
> We also provide a variant solution that estimates unknown intrinsics (with the initialization from GeoCalib, see table below and Appendix A.3). Our main setting follows the standard fixed-intrinsic assumption used in prior classical and 3DGS pipelines, as accurate intrinsics remain important for reconstruction. Extending the method to better handle unknown intrinsic is left for future work.
>
> | Methods| Tracking|PSNR|SSIM|LPIPS|
> | - | - | - | - | - |
> |Ours (w/o known intrinsics)|0.073|25.92|0.872|0.223|
> |Ours| 0.018|29.12|0.918|0.167|
>
> **W2:**
>
> Prior methods as Hierarchical 3D Gaussians and Octree-GS **assume complete global point clouds (offline COLMAP)** and do not directly transfer to **online, incremental** settings: LOD boundaries shift as new points arrive. We therefore introduce **a new pixel-wise, point-cloud-aware LOD rule** robust to incremental growth. For example, Octree-GS initializes its levels of detail (LOD) based on the nearest and farthest distances in the point cloud, but these distances dynamically change as the point cloud expands. To tackle the challenge, We replace LOD levels with a $d_{max}$ attribute for Gaussian selection. the distribution of the $d_{max}$ attribute is shown in figure 9 of the supplementary material. Additionally, we have conducted detailed ablation studies on the LOD design and the Gaussian selection mechanism.
>
> | Method| PSNR  | SSIM  | LPIPS | #Render(K) | MEM(MB) |
> | -- | - | - | - | - | - |
> | Ours| 29.12 | 0.918 | 0.167 | 130| 127.2   |
> | Ours w/o gs selection    | 29.12 | 0.918 | 0.167 | 176| 127.2   |
> | Ours w/o level-of-detail | 28.13 | 0.912 | 0.180 | 181| 133.3   |
>
> Furthermore, we conduct a multi-resolution experiment on the mipnerf360 dataset by following the setup of Octree-GS. We modify the Gaussian selection strategy as follows. We define $d_{rs} = s * d_r$, where $s$ is the scale factor obtained by dividing the standard resolution by the current resolution. During rendering, a Gaussian is included if $d_{rs} ≤ d_{max}$, excluded if $d_{rs} > 2*d_{max}$.
>
> The results are shown in the table below and figure 10 of the supplementary material. The introduction of anti-aliasing can significantly reduce the required Gaussian in the rendering of low-resolution images, while ensuring the rendering quality. Meanwhile, without introducing antilias, the body of the bicycle will become significantly thicker at low resolution, while the spokes will be thinner at high resolution.
>
> | **bicycle**| x1||| x1/2||| x1/4|| | x1/8  |||
> | ------------------ | ----- | ----- | ------- | ----- | ----- | ------- | ----- | ----- | ------- | ----- | ----- | ------- |
> | Method | PSNR  | SSIM  | #Render | PSNR  | SSIM  | #Render | PSNR  | SSIM  | #Render | PSNR  | SSIM  | #Render |
> | Ours w/o antialias | 20.27 | 0.397 | 1083K   | 21.67 | 0.475 | 1085K   | 23.60 | 0.614 | 1097K   | 25.01 | 0,743 | 1131K   |
> | Ours | 20.48 | 0.417 | 1008K   | 21.83 | 0.491 | 657K    | 23.62 | 0.621 | 167K    | 24.61 | 0.732 | 29K     |

---

> ### Author Response · Authors · 2025-11-24
> **Point-to-Point Response to Reviewer aQRV (2/2)**
>
> **W3:**
>
> Gaussian init/densification strategy is NOT claimed as a main contribution. Our contribution lies in **multi-resolution initialization tightly coupled with our new LOD and structured Gaussians**, enabling stable online reconstruction.
>
> **W4:**
>
> Thank you for bringing this perspective to our attention. We have carefully reviewed the related work you mentioned , and we respectfully point out the key differences between our loop-closure mechanism and CLIP-based loop detection. As Section 3.3 Loop Detection shows, CLIP still extracts **2D visual features** and performs **feature-similarity matching**, which remains an implicit encoding of scene overlap. In contrast, as described in Sec. 3.2 of our paper, $\pi^{3}$-based loop closure infers the corresponding point clouds of all relevant pixels **in a unified coordinate frame**, allowing frame-to-frame overlap to be obtained naturally through **explicit 3D point-cloud overlap**. Unlike similarity computation in 2D feature space, this direct 3D overlap estimation is substantially more accurate—an observation that is also confirmed by our ablation studies.
>
> Regarding the performance degradation when replacing $\pi^{3}$ with VGGT, the main reason is that VGGT tends to _hallucinate a viewpoint even when there is no actual overlap_ between frames (see Fig. 3 in the VGGT paper). This behavior can easily introduce incorrect loop-closure hypotheses for BA. Once a VGGT-based loop closure is wrong, all subsequent pose estimates become conditioned on this erroneous frame, causing a global drift and ultimately leading to poor evaluation results. However, if we segment the trajectory at the point where the incorrect loop closure occurs and evaluate the two sub-trajectories separately, the performance gap is actually small (This is the scene 126d on the table below). We observe that $\pi^{3}$ behaves more conservatively in non-overlapping viewpoints, which is precisely why we choose $\pi^{3}$ for our loop-closure module.
>
> | Methods            | 0077  | 02f2  | 0b03  | 126d  | 1cbb  | 2284  | 2d2e  | 3037  | 41eb  | 4600  | 4808  | 5462  | 712d  | 7543  |
> | ------------------ | ----- | ----- | ----- | ----- | ----- | ----- | ----- | ----- | ----- | ----- | ----- | ----- | ----- | ----- |
> | Ours w/ Loop(vggt) | 0.010 | 0.011 | 0.014 | 1.176 | 0.014 | 0.022 | 0.010 | 0.007 | 0.030 | 0.075 | 0.022 | 0.032 | 0.013 | 0.011 |
> | Ours               | 0.009 | 0.011 | 0.015 | 0.019 | 0.015 | 0.012 | 0.010 | 0.005 | 0.017 | 0.060 | 0.021 | 0.030 | 0.014 | 0.011 |
>
>
> To be honest, different foundation models indeed provide varying levels of information to BA, some are more robust, others less so, which naturally affects BA performance. Nevertheless, we believe that using a foundation model for loop closure, compared to 2D feature-based methods, offers a more intuitive and straightforward explanation and insight. Importantly, it can achieve superior results with much lower computational complexity than traditional feature-clustering-based approaches.
>
> Using $\pi^{3}$ does introduce some GPU memory overhead, but compared to the memory footprint of Gaussians (significantly influenced by the scale of scene), the overhead from $\pi^{3}$ is relatively modest. Compared to the overhead introduced by $\pi^{3}$, the intuitive and accurate loop closure detection it provides offers greater benefits.
>
> | Metric          | w/ $\pi^{3}$ | w/o $\pi^{3}$ |
> | --------------- | ------------ | ------------- |
> | GPU Peak Memory | 25.49GB      | 26.81GB       |
>
> [1] Lan, T., Lin, Q., & Wang, H. (2024). Monocular gaussian slam with language extended loop closure. _arXiv preprint arXiv:2405.13748_.
>
> [2] Wang, J., Chen, M., Karaev, N., Vedaldi, A., Rupprecht, C., & Novotny, D. (2025). Vggt: Visual geometry grounded transformer. In _Proceedings of the Computer Vision and Pattern Recognition Conference_ (pp. 5294-5306).
>
> **W5:**
>
> Thank you for your careful review. We have removed the duplicate references in the revised version.

---

> > ### Comment · Reviewer_aQRV · 2025-11-27
> >
> > I appreciate the detailed explanation and the additional experiments presented in the authors’ reply. After reading all rebuttal material, one further question regarding the choice of MASt3R and Pi3, (i.e., use different models for pose estimation and loop closure) was clarified in the response to Reviewer Wwfj. Overall, my concerns are sufficiently addressed and I will raise my score accordingly.

---

### Official Review · Reviewer_SdRB · 2025-10-30

**Soundness:** 3
**Presentation:** 3
**Contribution:** 3
**Rating:** 6
**Confidence:** 3

**Summary:**

I am not an expert in 3d vision and reconstruction, so the following comments and questions could be based on potentially inaccurate understandings or biased perspectives. From my understanding, this paper presents a system for efficient and high-fidelity 3D reconstruction performed "on-the-fly" from monocular image sequences. The method utilizes a \textit{structured scene representation} to achieve both speed and quality, demonstrating competitive performance against state-of-the-art SLAM and reconstruction approaches like DROID-SLAM and Go-SLAM regarding metrics like ATE and memory consumption. The goal is to deliver interactive, robust 3D reconstruction in real-time scenarios.

**Strengths:**

1. Achieves a desirable balance between reconstruction fidelity (low ATE/RTE, good PSNR) and computational efficiency (low time and memory footprint), crucial for real-time mobile applications.

2. The system's design is robust for on-the-fly operation and across diverse scene types (indoor and outdoor). The results show strong quantitative improvements or parity with several established recent baselines.

**Weaknesses:**

1. The abstract and snippets are light on the explicit technical details of the "structured scene representation." Without knowing the specific structure (e.g., hash grid, implicit surface, etc.), it is difficult to judge the approach's novelty and engineering complexity.

2. The comparison in the snippet only covers a limited subset of recent SLAM/Reconstruction methods; a more comprehensive benchmark, including performance on texture-less regions or under extreme motion, would be beneficial.

**Questions:**

Could the authors elaborate on the specific architecture of the "structured scene representation" and quantify how its design choices (e.g., sparsity, resolution, memory layout) explicitly contribute to the claimed efficiency gain compared to unstructured or purely implicit representations?

---

> ### Author Response · Authors · 2025-11-24
> **Point-to-Point Response to Reviewer SdRB (1/2)**
>
> We sincerely thank Reviewer SdRB for the comments on the content and experimental sections of the paper. These comments have made our paper more solid and logically coherent. Below are our point-by-point responses:
>
> **W1:**
>
> Thank you for your suggestion. We have added the technical details of the structured scene representation in the main text.
>
> A detailed explanation is provided below: The structure is **a spatially sparse grid attached to Gaussians**, decoding scale/rotation attributes jointly from grid features (`global_feat`) + Gaussian features (`local_feat`). This **enforces local consistency** and **accelerates convergence**.
>
> **Q1:**
>
> We have provided an explanation of the structural Gaussians in W1 and also conducted an new ablation study on the `global_feat` and `local_feat`. Results show consistent gains in PSNR/LPIPS with negligible memory overhead, verifying its necessity.
>
> | Method (Ours w/o level-of-detail)  | PSNR      | SSIM      | LPIPS     | #Render(K) | MEM(MB)   |
> | ---------------------------------- | --------- | --------- | --------- | ---------- | --------- |
> | w/o global_feat                    | 27.95     | 0.910     | 0.197     | 184        | 161.5     |
> | global_feat (8) + local_feat (24)  | 28.11     | 0.912     | 0.181     | 184        | 141.2     |
> | global_feat (16) + local_feat (16) | **28.13** | **0.912** | **0.180** | **181**    | **133.3** |
> | global_feat (24) + local_feat (8)  | 28.03     | 0.911     | 0.185     | 182        | 127.6     |
> | w/o local_feat                     | 27.98     | 0.910     | 0.191     | 183        | 121.7     |

---

> ### Author Response · Authors · 2025-11-24
> **Point-to-Point Response to Reviewer SdRB (2/2)**
>
> **W2:**
>
> Thank you for the suggestion, which indeed helps make our paper more comprehensive. We currently compare against several classic SLAM methods, including ORB-SLAM3 (TRO 2021), DROID-SLAM (NeurIPS 2021), Go-SLAM(ICCV 2023), DPV-SLAM++ (NeurIPS 2023; ECCV 2024), and MASt3R-SLAM (CVPR 2025). For 3DGS-based SLAM, we include MonoGS (CVPR 2024), S3PO-GS (ICCV 2025), SEGS-SLAM (ICCV 2025). For reconstruction methods, we compare with OnTheFly-NVS (TOG 2025) and LongSplat (ICCV 2025).
>
> **For new baselines:** We have added comparisons with the SLAM method LDSO (IROS 2018), the reconstruction method NICER-SLAM (3DV 2024) and feed-forward method AnySplat (TOG 2025). These methods, including the traditional approaches and the current state-of-the-art methods, make our experiments more comprehensive and complete.
>
> **For new datasets:** We further evaluate our system on the TUM RGB-D Structure vs. Texture category to assess performance in low-texture environments, and on TartanAirV2 Hard sequences to test robustness under extreme motion. In the supplementary material, we also provide videos recorded on our own sequences exhibiting extreme motion and severe low-texture conditions.
>
> (/ indicates the system crashed during operation.)
>
> **ATE RMSE Evaluation Results**
>
> | Method      | TUM (no texture) | TartanAirV2 (extreme motion) |
> | ----------- | ------------------ | ---------------------------- |
> | LDSO        | 0.207              | 0.230                        |
> | ORB-SLAM3   | /                  | /                            |
> | Go-SLAM     | 0.371              | 0.258                        |
> | DROID-SLAM  | 0.407              | 0.301                        |
> | DPV-SLAM    | **0.054**          | *0.223*                 |
> | MASt3R-SLAM | 0.076              | 0.537                        |
> | NICER-SLAM  | /                  | 1.666                        |
> | MonoGS      | 0.671              | /                            |
> | S3PO-GS     | 0.524              | 1.260                        |
> | SEGS-SLAM   | /                  | /                            |
> | OnTheFly    | 0.565              | 1.116                        |
> | LongSplat   | 0.231              | 1.244                        |
> | Ours        | *0.072*       | **0.095**                    |
>
> **Reconstruction Evaluation Results**
>
> | Dataset    | TUM (no texture) |           |           | TartanAirV2 (extreme motion) |           |           |
> | ---------- | ------------------ | --------- | --------- | ---------------------------- | --------- | --------- |
> | Method     | PSNR               | SSIM      | LPIPS     | PSNR                         | SSIM      | LPIPS     |
> | NICER-SLAM | /                  | /         | /         | 12.43                        | 0.445     | 0.671     |
> | MonoGS     | 27.43              | 0.912     | 0.353     | /                            | /         | /         |
> | S3PO-GS    | 28.97              | 0.916     | 0.340     | 14.87                        | 0.598     | 0.631     |
> | SEGS-SLAM  | /                  | /         | /         | /                            | /         | /         |
> | OnTheFly   | 26.62              | 0.918     | 0.355     | 15.34                        | 0.635     | 0.527     |
> | LongSplat  | 31.68              | 0.911     | 0.296     | 17.26                        | 0.639     | 0.500     |
> | Ours       | **33.74**          | **0.950** | **0.264** | **23.17**                    | **0.813** | **0.268** |
>
>
> **Special Evaluation with Feed-Forward Method**
>
> | Dataset            | ScanNet++ (in-domain) |           |           |           | Fast-livo2 (out-domain) |           |           |           |
> | ------------------ | --------------------- | --------- | --------- | --------- | ----------------------- | --------- | --------- | --------- |
> | Method             | Tracking              | PSNR      | SSIM      | LPIPS     | Tracking                | PSNR      | SSIM      | LPIPS     |
> | Anysplat (32view)  | 0.082                 | 23.41     | 0.822     | **0.176** | 0.235                   | 24.21     | 0.772     | 0.198     |
> | Ours (32view)      | **0.034**             | **26.53** | **0.856** | 0.201     | **0.065**               | **27.96** | **0.853** | **0.187** |
> | Anysplat (100view) | 0.128                 | 21.37     | 0.780     | 0.227     | 0.426                   | 21.28     | 0.672     | 0.287     |
> | Ours (100view)     | **0.076**             | **24.36** | **0.841** | **0.218** | **0.270**               | **24.56** | **0.788** | **0.255** |
>
>
> In fact, the strong performance of our system under low-texture conditions and extreme motion primarily stems from the robustness of the data-driven 3D foundation model. Our original motivation in designing this system was to support reliable 3D reconstruction in everyday indoor and outdoor environments. If you could recommend additional baselines or datasets that you believe we may have missed, we would greatly appreciate it.

---

### Official Review · Reviewer_Xx2g · 2025-11-02

**Soundness:** 3
**Presentation:** 3
**Contribution:** 2
**Rating:** 6
**Confidence:** 3

**Summary:**

ARTDECO provide a framework for real-time 3D reconstruction from monocular image sequences. ARTDECO aims to bridge the gap between high-fidelity per-scene optimization and feed-forward foundation models. ARTDECO achieves this by combining the efficiency of feed-forward inference with the accuracy and consistency of SLAM-based optimization, making it suitable for AR/VR, robotics, and digital twin applications.

**Strengths:**

Strength:
1. The paper is well-written and easy to follow.
2. ARTDECO achieve good result on the serveral benchmarks against recent baselines, providing a solid solution for 3D reconstruction

**Weaknesses:**

Major Weakness:
1. The overall pipeline involves multiple stages and components, making it appear rather complex.

Minor Weakness:
1. In line 176, “front-end” should be changed to “frontend” for consistency with the rest of the paper.

**Questions:**

Question:
1. How does ARTDECO compare with simpler baselines, such as $\pi^{3}$ combined with bundle adjustment?
2. In line 313, should the formula $\alpha = (d_r - d_{max})/ (d_{max})$ given that it occurs in the rendering process?
3. Can you provide more details on the ablation studies for mapper frames, level of detail, and structural Gaussians regarding the baseline and variant designs used?

---

> ### Author Response · Authors · 2025-11-24
> **Point-to-Point Response to Reviewer Xx2g (1/2)**
>
> We sincerely thank Reviewer Xx2g, whose comments on system design, ablation experiments, and typos have made our paper more complete and professional. Below are our point-by-point responses:
>
> **Major Weakness:**
>
> Our system is **not unnecessarily complex, **but is a **carefully integrated, functionally complete framework** designed for robustness and efficiency in long-sequence, streaming reconstruction. The framework contains only two core components:  (1) **a localization module** following the standard frontend/backend structure of modern**SLAM**, and (2)**a mapping module** adhering to the **3DGS** pipeline. We built upon these two techniques, **adding the essential modifications** required to make them operate reliably in an online setting.
>
> In contrast, recent feed-forward approaches (e.g., AnySplat) achieve simple framework but face **fundamental limitations**: they operate at **lower resolution**, exhibit **high memory usage** on long sequences, and lack the backend optimization needed to refine poses and predictions over time. As our quantitative comparisons on ScanNet++ and Fast-LiVO2 show, these limitations translate directly into **significant drops in both tracking accuracy and reconstruction fidelity**.
>
> Overall, the empirical evidence confirms that **a modular SLAM-style system with systematic post-optimization is necessary** for stable long-sequence performance, and that our design yields **substantial and consistent gains** over feed-forward alternatives.
>
> | Dataset | ScanNet++ | (in-domain) | | | Fast-livo2 | (out-domain) | | |
> |---------|----------|------|------|-------|----------|------|------|-------|
> | Methods | Tracking | PSNR | SSIM | LPIPS | Tracking | PSNR | SSIM | LPIPS |
> | Anysplat (32view) | 0.082 | 23.41 | 0.822 | 0.176 | 0.235 | 24.21 | 0.772 | 0.198 |
> | Ours (32view) | 0.034 | 26.53 | 0.856 | 0.201 | 0.065 | 27.96 | 0.853 | 0.187 |
> | Anysplat (100view) | 0.128 | 21.37 | 0.780 | 0.227 | 0.426 | 21.28 | 0.672 | 0.287 |
> | Ours (100view) | 0.076 | 24.36 | 0.841 | 0.218 | 0.270 | 24.56 | 0.788 | 0.255 |
>
> [1] Jiang, L., Mao, Y., Xu, L., Lu, T., Ren, K., Jin, Y., ... & Dai, B. (2025). AnySplat: Feed-forward 3D Gaussian Splatting from Unconstrained Views. TOG, _2505.23716_.
>
> **Minor Weakness:**
>
> Thank you for your reminder. We have modified this detail in the revision version.

---

> ### Author Response · Authors · 2025-11-24
> **Point-to-Point Response to Reviewer Xx2g (2/2)**
>
> **Q1:**
>
> We did evaluate $\pi^{3}$+BA. Our results show that **$\pi^{3}$+BA is fundamentally unsuitable for long-sequence streaming**, even with additional engineering. $\pi^{3}$ was designed for **multi-frame inference**, which causes **proportion distortion and severe layering artifacts** whenever viewpoints vary; these artifacts **persist even after metric-scale alignment** and accumulate aggressively over long sequences. To test this fairly, we implemented a **full long-sequence $\pi^{3}$-based system** (details in Appendix A.3). As shown in the table below, its performance is **substantially worse** than ours. Moreover, $\pi^{3}$ **does not include a pixel-correspondence head**, which is crucial for accurate **correspondence **refinement and pose estimation. MASt3R-SLAM’s ablation (Table 4) also demonstrates that removing correspondence features degrades ATE RMSE.
>
> | Methods      | Tracking | PSNR  | SSIM  | LPIPS |
> | ------------ | -------- | ----- | ----- | ----- |
> | $\pi^{3}$+BA | 0.374    | 15.46 | 0.713 | 0.438 |
> | Ours         | 0.018    | 29.12 | 0.918 | 0.167 |
>
> | Method                            | ATE RMSE |
> | --------------------------------- | -------- |
> | Point Matching                    | 0.062    |
> | Point Matching + Feature Matching | 0.039    |
>
>
> [1] Murai, R., Dexheimer, E., & Davison, A. J. (2025). MASt3R-SLAM: Real-time dense SLAM with 3D reconstruction priors. In _Proceedings of the Computer Vision and Pattern Recognition Conference_ (pp. 16695-16705).
>
> [2] Wang, Y., Zhou, J., Zhu, H., Chang, W., Zhou, Y., Li, Z., ... & He, T. (2025). $\pi^ 3$: Permutation-Equivariant Visual Geometry Learning. _arXiv preprint arXiv:2507.13347_.
>
> **Q2:**
>
> Thank you for your careful reminder. Yes, it should be $d_r$ here, and we have revised the original formula $\alpha = (d - d_{\max}) / d_{\max}$.
>
> **Q3:**
>
> We provide full ablations isolating the contribution of each component.
>
> **(1) Mapper frames.** Removing mapper frames forces Gaussian initialization to rely solely on keyframes, cutting initialization frames by **50–70%**. This produces **overly sparse geometry**, weaker coverage, and a **2.74 dB PSNR drop**, confirming that mapper frames are essential for maintaining density and stability in online mapping.
>
> **(2) Level-of-Detail (LoD).** Without LoD, Gaussians are initialized only at a single resolution and _all_ are rendered at all scales, increasing overhead and degrading both high- and low-frequency fidelity. Our LoD design yields **significant reduction in rendered Gaussians** while improving PSNR/SSIM, which is demonstrated in the ablation experiments below.
>
> **(3) Structural Gaussians.** Our structured design initializes a sparse grid around each Gaussian and decode scale/rotation attributes jointly from grid features (`global_feat`) + Gaussian features (`local_feat`). This enforces **local structural coherence** and **accelerates convergence**. To demonstrate the impact of structuring on reconstruction, we conduct a more detailed comparative experiment on `global_feat` and `local_feat`. Experiments show gains in PSNR/LPIPS with small memory overhead.
>
> | Method                               | PSNR  | SSIM  | LPIPS | #Render(K) | MEM(MB) |
> | ------------------------------------ | ----- | ----- | ----- | ---------- | ------- |
> | Ours                                 | 29.12 | 0.918 | 0.167 | 130        | 127.2   |
> | Ours w/o mapper frames               | 26.38 | 0.898 | 0.229 | 73         | 78.8    |
> | **Ours w/o level-of-detail**         |       |       |       |            |         |
> | - w/o global_feat                    | 27.95 | 0.910 | 0.197 | 184        | 161.5   |
> | - global_feat (8) + local_feat (24)  | 28.11 | 0.912 | 0.181 | 184        | 141.2   |
> | - global_feat (16) + local_feat (16) | 28.13 | 0.912 | 0.180 | 181        | 133.3   |
> | - global_feat (24) + local_feat (8)  | 28.03 | 0.911 | 0.185 | 182        | 127.6   |
> | - w/o local_feat                     | 27.98 | 0.910 | 0.191 | 183        | 121.7   |

---

### Author Response · Authors · 2025-12-01
**Rebuttal Summary**

**We deeply appreciate the effort and time the Area Chair has dedicated to handling our submission under these special circumstances.  We are also grateful for the thoughtful and constructive feedback provided from all the reviewers. To facilitate the review process, we would like to briefly outline the review and our responses.**

> **Reviewer Xx2g**
>
>
> The reviewer requested additional analysis of system complexity, ablations on different Foundation Models and structured scene representations, and noted several typographical errors. In response, we conducted a complexity analysis across 14 ScanNet++ scenes, added the requested ablation studies, and corrected the minor errors.
>
> *Initial score: 6. (no further response received before the special circumstances arose)*
>
> **Reviewer SdRB**
>
>
> The reviewer asked for clearer technical details regarding our structured scene representation and suggested evaluations on extreme scenarios (low texture, extreme motion), along with new baselines. We revised the paper to clarify the technical design, evaluated our method on four TUM RGBD low-texture scenes and seven TartanAirV2 Hard scenes, and added three widely used baselines (13 total).
>
> *Initial score: 6. (no further response received before the special circumstances arose)*
>
>
> **Reviewer aQRV**
>
> The reviewer requested clarification or empirical justification for several core design choices, including the use of the pinhole camera model, the LoD scene design, Gaussian initialization, and the loop-closure backbone, in addition to noting minor typographical issues. We added detailed explanations and performed targeted ablation experiments on 14 ScanNet++ scenes.
>
> *Initial score: 4 → revised to 6 after rebuttal.*
>
>
> **Reviewer Wwfj**
>
> The reviewer suggested to provide more technical details and ablations, especially regarding the selected Foundation Model, comparisons with feed-forward methods, and ablations on both the SLAM and mapping modules. We added these details and conducted the requested ablations on 14 ScanNet++ scenes.
>
> *Initial score: 6 → revised to 8 after rebuttal.*
>

---

### Meta-Review · Area_Chair_zY3R · 2025-12-23

**Summary:**

This paper proposes ARTDECO, a framework for on-the-fly 3D reconstruction from monocular image sequences. It combines the efficiency of feed-forward models with the reliability of SLAM-based pipelines. 3D foundation models are used for pose estimation and point prediction, and a Gaussian decoder is designed to output structured 3D Gaussians. In addition, a hierarchical Gaussian representation with a LoD-aware rendering strategy is introduced to sustain both fidelity and efficiency at scale.
The concerns of the reviewers include lack of analysis on system complexity, inadequate technical details, and insufficient evaluation and comparison. The rebuttal addressed all of them.

**Reviewer Concerns:**

Reviewer Xx2g requested additional analysis on complexity, ablations on different components. Reviewer SdRB requested more technical details regarding the proposed structured scene representation and more evaluations on extreme scenarios. Reviewer aQRV requested clarification for the core design choices. Reviewer Wwfj requested more technical details and more ablations. The concerns above were addressed by the rebuttal.

**Reviewer Scores:**

The paper initially received 6(Xx2g), 6(SdRB), 4(aQRV) and 6(Wwfj). After rebuttal, Reviewer aQRV and Wwfj raised the ratings to 6 and 8, respectively.

---

### Decision · Program_Chairs · 2026-01-26

Accept (Poster)